**Exploring the impacts of unprecedented climate extremes on forest ecosystems: hypotheses to guide modeling and experimental studies**

Jennifer A. Holm[1,*], David M. Medvigy[2], Benjamin Smith[3,4], Jeffrey S. Dukes[5], Claus Beier[6], Mikhail Mishurov[3], Xiangtao Xu[7], Jeremy W. Lichstein[8], Craig D. Allen[9], Klaus S. Larsen[6], Yiqi Luo[10], Cari Ficken[11], William T. Pockman[12], William R.L. Anderegg[13], and Anja Rammig[14]

[1] Lawrence Berkeley National Laboratory, Berkeley, California, USA

[2] University of Notre Dame, Notre Dame, Indiana, USA

[3] Dept of Physical Geography and Ecosystem Science, Lund University, Lund, Sweden

[4] Hawkesbury Institute for the Environment, Western Sydney University, Penrith, NSW 2751, Australia

[5] Department of Forestry and Natural Resources and Biological Sciences, Purdue University, West Lafayette, Indiana, USA

[6] Department of Geosciences and Natural Resource Management, University of Copenhagen, Frederiksberg, Denmark

[7] Department of Ecology and Evolutionary Biology, Cornell University, Ithaca, New York, USA

[8] Department of Biology, University of Florida, Gainesville, Florida, USA

[9] U.S. Geological Survey, Fort Collins Science Center, New Mexico Landscapes Field Station, Los Alamos, New Mexico, USA

[10] Center for Ecosystem Science and Society, Department of Biological Sciences, Northern Arizona University, Flagstaff, Arizona, USA

[11] Department of Biology, University of Waterloo, Waterloo, Ontario, Canada

[12] Department of Biology, University of New Mexico, Albuquerque, New Mexico, USA

[13] School of Biological Sciences, University of Utah, Salt Lake City, Utah, USA

[14] Technical University of Munich, TUM School of Life Sciences Weihenstephan, Freising, Germany

*Correspondence to*: Jennifer Holm; 510-495-8083; jaholm@lbl.gov

Keywords: demographic modeling; mortality; drought; recovery; carbon cycle; nonstructural carbohydrate storage; plant hydraulics; dynamic vegetation

**Abstract**
Climatic extreme events are expected to occur more frequently in the future, increasing the
likelihood of unprecedented climate extremes (UCEs), or record-breaking events. UCEs, such as
extreme heatwaves and droughts, substantially affect ecosystem stability and carbon cycling by
increasing plant mortality and delaying ecosystem recovery. Quantitative knowledge of such
effects is limited due to the paucity of experiments focusing on extreme climatic events beyond
the range of historical experience. Here, we present a road map of how two dynamic vegetation
demographic models (VDMs) can be used to investigate hypotheses surrounding ecosystem
responses to UCEs (e.g., unprecedented droughts). As a result of nonlinear ecosystem responses
to UCEs, that are qualitatively different from responses to milder extremes, we consider both
biomass loss and recovery rates over time, by reporting a time-integrated carbon loss as a result
of UCE, relative to the absence of drought. Additionally, we explore how unprecedented
droughts in combination with increasing atmospheric $CO_2$ and/or temperature may affect
ecosystem stability and carbon cycling. We explored these questions using simulations of pre-
drought and post-drought conditions at well-studied forest sites, using the ED2 and LPJ-GUESS
models. Due to the two models having different but plausible representations of processes and
interactions, they diverge in sensitivity of nonlinear biomass loss due to drought duration or
intensity, and differ between each site. Biomass losses are most sensitive to drought duration in
ED2, but to drought intensity in LPJ-GUESS. Elevated atmospheric $CO_2$ concentrations ($eCO_2$)
alone did not completely buffer the ecosystems from carbon losses during UCEs in the majority
of our simulations. Our findings highlight contrasting differences in process formulations and
uncertainties in models, most notably related to availability in plant carbohydrate storage and the
diversity of plant hydraulic schemes, in projecting potential ecosystem responses to UCEs. We
provide a summary of the current state and role of many model processes that give way to
different underlying hypotheses of plant responses to UCEs, reflecting knowledge gaps, which
should be tested with targeted field experiments and an iterative modeling-experimental
conceptual framework.

## 1 Introduction

The increase in extreme climate and weather events, such as prolonged heatwaves and droughts as seen over the last three decades, are expected to continue to increase in frequency and magnitude, leading to progressively longer and warmer droughts on land (IPCC 2012, 2021). Droughts are affecting all areas of the globe, more than any other natural disturbance, and recent droughts have broken long-standing records (Ciais et al., 2005; Phillips et al., 2009; Williams et al., 2012; Matusick et al., 2013; Griffin and Anchukaitis, 2014; Asner et al., 2016; Feldpausch et al., 2016; Seneviratne et al., 2021). Such 'unprecedented climate extremes' (UCEs; "record-breaking events", IPCC (2012)) that are larger in extent and longer-lasting than historical norms can have dramatic consequences for terrestrial ecosystem processes, including carbon uptake and storage and other ecosystem services (Reichstein et al., 2013; Settele, 2014; Allen et al., 2015; Brando et al., 2019; Kannenberg et al., 2020). Thus, to better anticipate the implications of climatic changes for the terrestrial carbon sink and other ecosystem services, we need to better understand how ecosystems respond to extreme droughts and other UCEs.

To learn how ecosystems respond to rarely experienced or unprecedented conditions, ecologists can experimentally manipulate environmental conditions (Rustad, 2008; Beier et al., 2012; Meir et al., 2015; Aguirre et al., 2021). However, the majority of such experiments apply moderate treatments based on a historical sense, which are mostly weaker in intensity and/or shorter in duration than potential future UCEs (Beier et al., 2012; Kayler et al., 2015; but see Luo et al., 2017), and single experiments have low power to detect effects of stressors on ecosystem responses (Yang et al., 2022). Additionally, most experiments examine low-stature ecosystems, such as grassland, shrubland or tundra, due to lower requirements for infrastructure and financial investment compared to mature forests. However, forests may respond qualitatively differently to UCEs than other ecosystems, in part due to mortality of large trees and strong nonlinear ecosystem responses, with long-lasting consequences for ecosystem-climate feedbacks (Williams et al., 2014; Meir et al., 2015). Ecosystem responses to naturally occurring extreme droughts and heatwaves have been documented (Ciais et al., 2005; Breshears et al., 2009; Feldpausch et al., 2016; Matusick et al., 2016; Ruthrof et al., 2018; Powers et al,. 2020); however, these rapidly-mobilized post-hoc studies often are unable to measure all critical variables and may lack consistently collected data for comparison with pre-drought conditions, thus limiting their inferential power and ability to improve quantitative models. The difficulties of performing

controlled real-world experiments of UCEs at broad spatial and temporal scales make process-based modeling a valuable tool for studying potential ecosystem responses to extreme events.

Process-based models can be used to explore potential ecosystem impacts using projected climate change over broad spatial and temporal scales (Gerten et al., 2008; Luo et al., 2008; Zscheischler et al., 2014; Sippel et al., 2016), as seen in a few modeling studies that have synthesized and improved our process-level understanding of UCE effects (McDowell et al., 2013; Dietze and Matthes, 2014). However, due to the overly simplified representation of ecological processes in most land surface models (LSMs) – the terrestrial components of Earth System Models (ESMs) used for climate projections – it is doubtful whether most of these models adequately capture ecosystem feedbacks and other responses to UCEs (Fisher and Koven, 2020). For example, only a few ESMs in recent coupled model intercomparison projects (CMIP6) and IPCC climate assessments (Ciais et al., 2013; Arora et al., 2020) include vegetation demographics (Döscher et al., 2022), and most rely on prescribed, static maps of plant functional types (PFTs) (Ahlström et al., 2012). Other LSMs simulate PFT shifts (i.e., dynamic global vegetation models, DGVMs; Sitch et al., (2008)) based on bioclimatic limits, instead of emerging from the physiology- and competition-based demographic rates that determine resource competition and plant distributions in real ecosystems (Fisher et al., 2018). Although a new generation of LSMs with more explicit ecological dynamics and structured demography is emerging (Holm et al., 2020; Koven et al., 2020; Döscher et al., 2022), most current ESMs are limited in ecological detail and realism (e.g., ecosystem structure, demography, and disturbances). Failing to mechanistically represent mortality, recruitment, and disturbance – each of which influences biomass turnover and carbon (C) allocation (Friend et al., 2014) – limits the ability of these models to realistically forecast ecosystem responses to anomalous environmental conditions like UCEs (Fisher et al., 2018).

Evaluating and improving the representation of physiological and ecological processes in ecosystem models is critical for reducing model uncertainties when projecting the effects of UCEs on long-term ecosystem dynamics and functioning. Vegetation demography, plant hydraulics, enhanced representations of plant trait variation, explicit treatments of resource competition (e.g., height-structured competition for light), and representing major disturbances (e.g., extreme drought) have all been identified as critical areas for advancing current models (Scheiter et al., 2013; Fisher et al., 2015; Weng et al., 2015; Choat et al., 2018; Fisher et al.,

2018; Blyth et al., 2021) and are necessary advances for realistically representing the ecosystem
impacts of UCEs. In this perspectives focused paper we look at the differences in these
processes, and how they contribute to uncertainty across multiple temporal phases surrounding
an extreme event: predicting an ecosystem's pre-disturbance resistance, which influences the
degree of impact and recovery from UCEs. Table 1 describes a summary of model mechanisms
that affect pre-drought resistance and post-drought recovery and we suggest are critical areas
further research (ca. Frank et al., 2015).
In order to inform our discussion, we explore the potential responses of forest ecosystems
to UCEs using two state-of-the-art process-based demographic models (vegetation demographic
models, VDMs; Fisher et al., (2018)), a unique model exploration-discussion approach to help
highlight new paths forward for model advancement. We first present conceptual frameworks
and hypotheses on potential ecosystem responses to UCEs based on current knowledge. We then
present VDM simulations for a range of hypothetical UCE scenarios to illustrate current state-of-
the-art model representations of eco-physiological mechanisms expected to drive responses to
UCEs. While a variety of UCE-linked biophysical tree disturbance processes (e.g., fire, wind,
insect outbreaks) can drive nonlinear ecosystem responses, we focus specifically on extreme
droughts, which have important impacts on many ecosystems around the world (e.g. Frank et al.,
2015, IPCC 2021). By studying modeled responses to UCEs, we explore the limits to our current
understanding of ecosystem responses to extreme droughts and their corresponding thresholds
and tipping points. As anthropogenic forcing has increased the frequency, duration, and intensity
of droughts throughout the world (Chiang et al., 2021), we explore how $eCO_2$ and rising
temperatures may affect drought-induced C loss and recovery trajectories, and how the scientific
community can iteratively address these questions through experiments and modeling studies.
We believe the combination of using cutting-edge VDMs alongside an inspection of current gaps
in knowledge will help guide modeling and experimental advances in order to address novel
forest responses to climate extremes.

**1.1 Conceptual and Modeling Framework for Hypothesis Testing:**
We combine conceptual frameworks (Fig. 1) and ecosystem modeling to test two
hypotheses on potential responses of plant carbon stocks to UCEs. The first hypothesis is:
*Hypothesis (H1). Terrestrial ecosystem responses to UCEs will differ qualitatively from*
*ecosystem responses to milder extremes because responses are nonlinear. Nonlinearities can*
*arise from multiple mechanisms – including shifts in plant hydraulics, C allocation,*
*phenology, and stand demography – and can vary depending on the pre-drought state of the*
*ecosystem.*
We present four conceptual relationships that describe terrestrial ecosystem responses to varying
degrees of extreme events (Fig. 1). Change in vegetation C stock is related to drought intensity
and/or drought duration in a near-linear relationship (Fig. 1a, H0, null hypothesis), which has
some observational support from annual and perennial grassland ecosystems, shrublands and
savannas across the globe (Bai et al., 2008; Muldavin et al., 2008; Ruppert et al., 2015). We
recognize that most ecological systems are nonlinear, thus alternatives to the null hypothesis are
that biomass loss increases nonlinearly with increased drought intensity (i.e., reduction in
precipitation) represented by a threshold-based relationship (Fig. 1a, H1a), increased drought
duration (i.e., prolonged drought with the same intensity) by shifting the near-linear relationship
downwards via increasing slopes (Fig. 1a, H1b), or the combination of both intensity and
duration (Fig. 1a, H1c). These hypotheses are supported by observations from the Amazon Basin
and Borneo (Phillips et al., 2010) where tree mortality rates increased nonlinearly with drought
intensity. Similarly, plant hydraulic theories predict nonlinear damage to the plant-water
transport systems, and thus mortality risk, as a function of drought stress (Sperry and Love,
2015). In particular, longer droughts are more likely to lead to lower soil water potentials,
leading to a nonlinear xylem damage function even if stomata effectively limit water loss (Sperry
et al., 2016).
*Hypothesis (H2): The effects of increasing atmospheric $CO_2$ concentration ($eCO_2$) will*
*alleviate impacts of extreme drought stress through an increase in vegetation productivity and*
*water-use efficiency, but only up to a threshold of drought severity, while increased*
*temperature (and related water stress) will exacerbate tree mortality.*
This second hypothesis is based on growing evidence that effects of $eCO_2$ and climate
warming may interact with effects of drought intensity on ecosystems. The $CO_2$ fertilization
effect enhances vegetation productivity (e.g., net primary production, NPP) (Ainsworth and
Long, 2005; Norby et al., 2005; Wang et al., 2012), but this fertilization effect is generally
reduced by drought (Hovenden et al., 2014; Reich et al., 2014; Gray et al., 2016). Drought events
often coincide with increased temperature, which intensifies the impact of drought on
ecosystems (Allen et al., 2015; Liu et al., 2017), resulting in nonlinear responses in mortality
rates (Adams et al., 2009; Adams et al., 2017a). The evaluation of C cycling in VDMs with
doubling of $CO_2$ (only "beta effect") showed a large carbon sink in a tropical forest (Holm et al.,
2020), but the inclusion of climate interactions in VDMs needs to be further explored.
Here, we relate ecosystem responses to UCEs by calculating the "integrated carbon (C)
loss" (Fig. 1b and see Methods), which integrates C loss from the beginning of the drought until
the time when C stocks have recovered to 50% of the pre-drought level. In response to drought,
warming, and $eCO_2$, divergent potential C responses (gains and losses; Fig. 1c) can be expected
(Keenan et al., 2013; Zhu et al., 2016; Adams et al., 2017a). For example, a grassland
macrocosm experiment found that $eCO_2$ completely compensated for the negative impact of
extreme drought on net carbon uptake due to increased root growth and plant nitrogen uptake,
and led to enhanced post-drought recovery (Roy et al., 2016). However, a 16-year grassland
FACE and the SoyFACE experiments showed that $CO_2$ fertilization effects were reduced or
eliminated under hotter/drier conditions (Gray et al., 2016; Obermeier et al., 2016). Reich et al.,
(2014) also found that $CO_2$ fertilization effects were reduced in a perennial grassland by water
and nitrogen limitation.
A corollary to our H2 is that conditions that favor productivity (e.g., longer growing
seasons and/or $CO_2$ fertilization) will enhance vegetation growth leading to "structural
overshoot" (SO; Fig. 1d; adapted from and supported by Jump et al., 2017), and can amplify the
effects of UCEs. Enhanced vegetation growth coupled with environmental variability can lead to
exceptionally high plant-water-demand during extreme drought and water stress, resulting in a
"mortality overshoot" (MO; Fig 1d). We conceptualize how oscillations between SO and
associated MO could be amplified by increasing climatic variability and UCEs (Fig. 1d).
Confidence is low as to how historically unprecedented $eCO_2$ levels and temperatures will affect
ecosystems in the future (i.e., the widening of the shaded areas compared to historical, Fig. 1d).
We expect, however that a rapidly changing climate, combined with effects of UCEs as a result
of more frequent extreme drought/heat events and drought stress, can exacerbate and amplify
SOs and MOs (Jump et al., 2017), leading to increasing C loss, even though various buffering
mechanisms exist (cf. (Lloret et al., 2012; Allen et al., 2015)). Relative to our conceptual (Fig.
1d), we note that most experimental, observational and modeling studies (Ciais et al., 2005; da
Costa et al., 2010; Phillips et al., 2010; Meir et al., 2015) take into account only low to moderate
drought intensities (such as 50% rain excluded) or single events, or combine drought with
moderate effects of temperature change. Where there has been 100% rain exclusion, it was on
very small plots of 1.5 m$^2$ (Meir et al., 2015). As represented by the increasing amplitude of
oscillations in Fig. 1d, the interactions between increased temperatures, UCE events, and
vegetation feedbacks make ecosystem states become inherently unpredictable, particularly over
longer time-scales.

**224 2 Vegetation Demography Model (VDM) Approaches**

We argue that VDMs are well suited to address climate change impacts due to the
inclusion of detailed process representation of dynamic plant growth, recruitment, and mortality,
resulting in changes in abundance of different PFTs, as well as vertically stratified tree size- and
age-class structured ecosystem demography. Community dynamics and age-/size-structure are
emergent properties from competition for light, space, water, and nutrients, which dynamically
and explicitly scale up from the tree, to stand, to ecosystem level. Within this characterization,
VDMs also differ between each other and are set up in different configuration, allowing for
various testing capabilities. For full names of each model listed below and references, see Table
S1. For example, VDMs can aggregate and track the community level disturbance into either
patch-tiling sampling (e.g., ED2, FATES, LM3-PPA, ORCHIDEE, JSBACH4.0) or statistical
approximations (e.g., LPJ-GUESS, SEIB-DGVM, and CABLE-POP). VDMs could also vary in
representing light competition within either multiple canopy layers (e.g., ED2, FATES, LM3-
PPA, LPJ-GUESS, SEIB-DGVM) or in a single canopy (e.g., JSBACH4.0, ORCHIDEE,
CABLE-POP).
Powell et al. (2013) compared multiple VDMs and LSMs to interpret ecosystem
responses to long-term droughts in the Amazon and are informative when conducting model-data
comparisons, but studies of the cascade of ecosystem responses and mortality to UCEs are
lacking. In a cutting-edge area of development, new mechanistic implementation of plant
competition for water and plant hydraulics in VDMs (i.e., hydrodynamics) are improving our
understanding of plant-water relations and stresses within plants, such as with TFSv.1-Hydro
(Christoffersen et al., 2016), ED2-hydro (Xu et al., 2016), and FATES-HYDRO (Ma et al., 2021;
Fang et al., 2022). Compared to more simplistic representation of plant acquiring soil moisture
not connected to plant physiology (e.g., LPJ-GUESS, LM3-PPA, CABLE-POP, SEIB-DGVM).
For hydrodynamic representations in 'big-leaf' LSMs such as CLM5, JULES, and Noah-MP-
PHS see Kennedy et al., (2019), Eller et al., (2020), and Li et al., (2021) respectively.

The discussion section provides a deeper investigation of model response to UCEs related

to droughts. An exhaustive review of all VDMs, and all plant processes is too large to be done
here. Existing review papers of different VDM development, processes, and uncertainties can be
found here: Fisher et al., (2018); Bonan (2019); Trugman et al., (2019); Hanbury-Brown et al.
(2022); Bugmann and Seidl (2022); and specifically related to plant hydraulics see: Mencuccini
et al., (2019); Anderegg and Venturas (2020). We use LPJ-GUESS and ED2 as example VDMs
in an initial guide framework to explore hypotheses around vegetation mortality and integrated
carbon loss from UCEs and climate change impacts, and highlight limiting model processes.
Since field data needed to evaluate UCE responses are, by definition, unavailable, we do not
perform model-data comparisons. Rather, we use the model results and conceptual framework as
a road map to explore our hypotheses and illustrate their implications for ecosystem responses
under UCEs, not historical drought events.

**2.1 LPJ-GUESS and ED2 Model Descriptions**

We explored our hypotheses at forested ecosystems in Australia and Central America

using two VDMs: the Lund-Potsdam-Jena General Ecosystem Simulator (LPJ-GUESS) (Smith et
al., 2001; Smith et al., 2014) and the Ecosystem Demography model 2 (ED2) (Medvigy et al.,
2009; Medvigy and Moorcroft, 2012). Both LPJ-GUESS and ED2 resolve vegetation into tree
cohorts characterized by their PFT, in addition to age-class in LPJ-GUESS; and size, and stem
number density in ED2. Both models are driven by external environmental drivers (e.g.,
temperature, precipitation, solar radiation, atmospheric $CO_2$ concentration, nitrogen deposition),
and soil properties (soil texture, depth, etc.), and also depend on dynamic ecosystem state, which
includes light attenuation, soil moisture, and soil nutrient availability. Establishment and growth
of PFTs, and their carbon-, nitrogen- and water-cycles, are simulated across multiple patches per
grid cell to account for landscape heterogeneity. Both models characterize PFTs by physiological
and bioclimatic parameters, which vary between the models (Smith et al., 2001; Smith et al.,
2014; Medvigy et al., 2009; Medvigy and Moorcroft, 2012).

The LPJ-GUESS includes three woody PFTs: evergreen, intermediate evergreen, and

deciduous PFTs. Mortality in LPJ-GUESS is governed by a 'growth-efficiency'-based function
(kg C m$^{-2}$ leaf yr$^{-1}$), which captures effects of water deficit, shading, heat stress, and tree size on
plant productivity relative to its resource-uptake capacity (leaf area), with a threshold below
which stress-related mortality risk increases markedly, in addition to background senescence and
exogenous disturbances. Stress mortality can be reduced by plants using labile carbon storage,
modeled implicitly using a 'C debt' approach, which buffers low productivity, enhancing
resilience to milder extremes (more details are given in section 4.1.4). Total mortality can thus be
impacted by variation in environmental conditions such as water limitation, low light conditions,
and nutrient constraints, as well as current stand structure (Smith et al., 2001; Hickler et al.,

2004).

The ED2 version used here (Xu et al., 2016) includes four woody PFTs: evergreen,

intermediate evergreen, deciduous, brevi-deciduous, and deciduous stem-succulent. This ED2
version includes coupled photosynthesis, plant hydraulics, and soil hydraulic modules (Xu et al.,
2016), which together determine plant water stress. The plant hydraulics module tracks water
flow along a soil–plant–atmosphere continuum, connecting leaf water potential, stem sap flow,
and transpiration, thus influencing controls on photosynthetic capacity, stomatal closure,
phenology, and mortality. Leaf water potential depends on time-varying environmental
conditions as well as time-invariant PFT traits. Leaf shedding is triggered when leaf water
potential falls below the turgor loss point (a PFT trait) for a sufficient amount of time. Leaf
flushing occurs when stem water potential remains high (above half of the turgor loss point) for a
sufficient time (see Xu et al., 2016 for details). PFTs differ in their hydraulic traits, wood
density, specific leaf area, allometries, rooting depth, and other traits. Stress-based mortality in
the ED2 version used here includes two main physiological pathways in our current
understanding of drought mortality (McDowell et al., 2013): C starvation and hydraulic failure.
Mortality due to C starvation in ED2 results from a reduction of C storage, a proxy for non-
structural carbohydrate (NSC) storage, which integrates the balance of photosynthetic gain and
maintenance cost under different levels of light and moisture availability. Mortality due to
hydraulic failure in ED2 is based on the percentage loss of stem conductivity. ED2 also includes
a density-independent senescence mortality rate based on wood density.
**2.2 Modeling guide**

To exemplify how VDMs can be tools to explore new hypotheses related to UCEs we

applied the models at two field sites, that were chosen due to being extensively studied and the
models used here have already been run at these sites and previously benchmarked against field
data (see Xu et al., 2016; Medlyn et al., 2016; Medvigy et al., 2019 for model-data validation).
The purpose of this paper was not to do a large multi-site comparison, but rather just select a few
for hypothesis testing. In addition, the two sites span a range of vegetation types and are in
warm, seasonally dry climates that are more likely to experience droughts in the future (Allen et
al., 2017). The first is a mature *Eucalyptus* (*E. tereticornis*) warm temperate-subtropical
transitional forest that is the site of the Eucalyptus Free Air $CO_2$ enrichment (EucFACE)
experiment in Western Sydney, Australia (Medlyn et al., 2016; Ellsworth et al., 2017; Jiang et
al., 2020). The second site is a seasonally dry tropical forest in the Parque Nacional Palo Verde
in Costa Rica (Powers et al., 2009). Site description details can be found in Supplement Text A.

We performed a 100-year "baseline" simulation for each model at each site driven by

constant, near ambient, atmospheric $CO_2$ (400 ppm) and recycled historical site-specific climate
data (1992-2011 for EucFACE and 1970-2012 for Palo Verde; Sheffield et al., (2006)), absent of
drought treatments. A detailed description of the meteorological data and initial conditions used
to drive the models is in the Supplementary Text A. The two models were previously tuned for
each site (Xu et al., 2016; Medlyn et al., 2016), and no additional site-level parameter tuning was
conducted here due to evaluating responses from hypothetical UCEs. To describe the ecosystem
impact of UCEs, we simulated 10 years of pre-drought conditions (continuing from the baseline
simulation), followed by drought treatments that differed in intensity and duration, followed by a
100-year post-drought recovery period. To explore the effects of drought intensity, we conducted
20 different artificial drought intensity simulations, in which precipitation during the whole year
is reduced by 5% to 100% of its original amount, in increments of 5%. To explore the effects of
drought duration, the 20 different drought intensities are maintained over 1, 2 and 4 years (Table
S2). We examined model responses of aboveground biomass, leaf area index (LAI), stem density
(number ha$^{-1}$), plant available soil water (mm), plant C storage (kg C m$^{-2}$), change in stem
mortality rate (yr$^{-1}$), and PFT composition.

To explore how temperature, eCO$_2$ concentration, and UCE droughts influence forest C dynamics individually and in combination, we implemented the following five experimental scenarios, some realistic and others hypothetical, for each model (Table S2): increased temperature only (+2K over ambient), eCO$_2$ only (600 ppm and 800 ppm), and both increased temperature and eCO$_2$ (+2K 600 ppm; +2K 800 ppm). Temperature and eCO$_2$ manipulations were applied as step increases over the baseline conditions, and are artificial scenarios, as opposed to model-generated climate projections.

**2.3 Linking concepts, hypotheses, and model outcomes**

To relate our simulation results to Fig. 1a, we compared the total biomass loss as a result of each drought treatment by calculating the percentage of biomass reduction at the end of the drought period relative to the baseline (no drought) simulation. To explicitly consider biomass recovery rates over time, we calculated "integrated-C-loss" (Eqs. 1-3), as a result of drought under current climate, which are determined based on the concepts in Fig. 1b. We defined "integrated-C-loss" as the time-integrated carbon in biomass that is lost due to drought relative to what the vegetation would have stored in the absence of drought. That is, it is the difference between biomass in the presence of drought (B$_d$) at time (t) and biomass in the baseline simulation (no drought; B$_{base}$), integrated over a defined recovery time period (in kg C m$^{-2}$ yr):

$$\text{Integrated-C-loss} = \int_{t=t_1}^{t=t_2} (B_{base}(t) - B_d(t))\, dt$$

(Eq. 1)

To define the bounds of integration, in Eq. 1, $t_1$ is defined as the time when the maximum amount of plant C is lost as a result of the drought:

$$B_{base}(t_1) - B_d(t_1) = \max_t [B_{base}(t) - B_d(t)]$$

(Eq. 2)

Then, $t_2$ is defined implicitly as the time when 50% of the lost biomass has been recovered compared to the baseline:

$$B_{base}(t_2) - B_d(t_2) = \frac{1}{2}(B_{base}(t_1) - B_d(t_1))$$

(Eq. 3)

Since all integrated-C-loss results are taken as the difference from a non-drought baseline
biomass ($B_{base}$) and all droughts will result in a loss of C.

We also use integrated-C-loss to examine the role of drought, temperature and $eCO_2$

change for moderating or exacerbating the impacts of drought on forest C stocks; i.e., to evaluate
the hypotheses illustrated in Fig. 1c. To assess these impacts of changing climates, we calculate
an "integrated-C-change" (Eq. 4). Defined as the difference between the integrated-C-loss due to
drought alone (Eqs. 1-3) under present climate, and the integrated-C-loss due to the combined
effects of drought and climate change (i.e., five scenarios of temperature increase and $eCO_2$):

$$\text{Integrated-C-change} = integrated\ C\ Loss_{Drought} - integrated\ C\ Loss_{Drought+CC}$$

(Eq. 4)

Because we expect drought to reduce vegetation C stocks, and thus integrated-C-loss to

be negative, positive values of integrated-C-change indicate that changes in climatic drivers
reduced the C losses from drought (i.e., buffering effects). Negative values of integrated-C-
change indicate that the climate change scenario leads to either greater C losses or losses that
persist for longer amounts of time (i.e., magnitude and/or duration) compared to a simulation
with no climate change (i.e., "reference" run).

**3 Results**

As a basis for the treatment results presented here, we compared the baseline simulations

(prior to drought or climate change treatments) of the two VDMs to observations at both sites for
biomass and LAI (Table S3, Fig. S1). Both models had similar biomass compared to
observations at Palo Verde (10.4 - 11.7 vs. 11.0 kgC m$^{-2}$), and at EucFACE biomass matched
well in LPJ-GUESS (12.1 vs. 12.7 kgC m$^{-2}$) but was low in ED2 (5.6 kgC m$^{-2}$). Both models also
had similar LAI to observations at Palo Verde (3.3 – 4.5 vs. 3.8 ($\pm$ 1.06) m$^2$ m$^{-2}$), and at
EucFACE LAI matched well in ED2 (1.6 vs. 1.7 m$^2$ m$^{-2}$), but was high for LPJ-GUESS (3.2 m$^2$
m$^{-2}$). At EucFACE LAI ranged from 1.2 to 2.1 over a 28-month measurement period (Duursma
et al., (2016), but LPJ-GUESS had very large fluctuations in annual LAI outside of these ranges
(Fig. S1). These models are well documented and investigated VDMs, with many studies that
have looked into parameter uncertainty (see Supplemental Text A for select references that
explore model/parameter sensitivity).
Both models displayed nonlinear responses to drought, in concurrence with Hypothesis
H1, but they differ in their behavior and between sites. In general, ED2 shows sensitivity to
drought duration (Hypothesis H1b), while LPJ-GUESS shows a stronger sensitivity to drought
intensity (Hypothesis H1a). ED2's sensitivity to the duration of drought was mild at Palo Verde
(Fig. 2a), and stronger at EucFACE particularly during the 4-year drought with a strong non-
monotonic pattern (see explanation below) (Fig. 2b). When reporting only percentage of biomass
loss, ED2 predicts close to no UCE response at Palo Verde; with a maximum biomass reduction
of only 40% during 95% precipitation removal and a 4-year drought event (i.e., UCE). LPJ-
GUESS shows very little sensitivity to drought duration but is highly sensitive to drought
intensity. C loss predicted by LPJ-GUESS at Palo Verde reached a threshold at ~65% drought
intensity, after which forests exhibit strong biomass losses, up to 100% (Fig. 2a). At the
EucFACE site, both models predict a critical threshold of biomass loss at 35%-45% drought
intensity, with LPJ-GUESS predicting total biomass loss (up to 100%) after this drought
intensity threshold (Fig. 2b). The EucFACE drought threshold is lower than that of the
seasonally dry mixed tropical forest in Palo Verde.
With respect to C loss over a recovering time period (integrated-C-loss), the two models
predict similar drought responses at Palo Verde (Fig. 2c), but not at EucFACE (Fig. 2d). At Palo
Verde, the similarity between models in integrated-C-loss reflected longer biomass recovery time
but less biomass loss in the short-term in ED2 relative to LPJ-GUESS, which predicted greater
biomass loss immediately after drought but shorter recovery time. With the exception of the 1-
year drought in ED2, both models predict similar integrated-C-loss across a range of UCEs at
Palo Verde, via different pathways. The integrated-C-loss metric revealed a strong nonlinear
response to drought duration in ED2 (Fig. 2c), while this nonlinearity is less evident when only
examining change in biomass (Fig. 2a). The "V"-shaped patterns observed particularly in Fig.
2b, arise from interactions between whole-leaf phenology and stomatal responses to drought in
ED2. For drought intensities lower than 40%, stomatal conductance is reduced but leaves are not
fully shed. Leaf respiration continues, gradually depleting non-structural C pools, followed by a
loss of biomass. However, for higher drought intensities, leaf water potentials quickly become
systematically lower than leaf turgor loss points and tree cohorts shed all their leaves. This
strategy represents an immediate loss of C via leaf shedding, but spares the cohort from slow,
respiration-driven depletion of C stocks.

## 3.1 Predicted model responses to UCE droughts combined with increased temperature and/or eCO$_2$

Relating to our second hypothesis of additional effects of warming and eCO$_2$, we tested 15 treatments in total, repeating the five climate change scenarios for each of the three drought durations. With the addition of climate change impacts, ED2 remained sensitive to the duration of drought, with warming negatively impacting integrated-C-change and most consistently during 2- and 4-year drought durations. ED2 predicts that during the 2- and 4-year droughts at EucFACE, losses are exacerbated when accompanied with warming, even with eCO$_2$, with 600 ppm having a more detrimental impact than the more elevated 800 ppm (Fig. 3b-c). The average integrated-C-change was -111.0 kg C m$^{-2}$ yr across all 15 treatments (Table 2). Only during the 1-year drought duration did drought plus warming and eCO$_2$ have a buffering effect on C stocks, seen in four out of our five scenarios but only during relatively modest droughts intensities (Fig. 3a; i.e., positive integrated-C-change, see also Table 2).

The ED2 simulations of the seasonally dry Palo Verde site (Fig. 3d-f), produced less frequent negative impacts on drought and climate change driven C losses compared to EucFACE, with an average integrated-C-change of -53.9 kg C m$^{-2}$ yr$^{-1}$ across all 15 treatments (Table 2). During the 2-year drought, applying +2K with eCO$_2$ to 600 ppm showed a slight buffering effect to droughts and the most consistent positive integrated-C-change (Fig. 3e; Table 2). Interestingly, an increase in only eCO$_2$ to 800 ppm (no warming) when applied with the 2- and 4-year droughts resulted in the largest loss in integrated-C-change (Fig. 3e-f), larger than the expected 'most severe' scenario; +2K and 800 ppm.

Similar to ED2, the LPJ-GUESS model showed a nearly complete negative response in integrated-C-change as a result of UCE drought and scenarios of warming and eCO$_2$ at the EucFACE site (Fig. 3g-i), but mixed and more muted results at Palo Verde (Fig. 3j-l, Table 2). The average integrated-C-change relative to the reference case was -95.4 at EucFACE and -7.8 kg C m$^{-2}$ yr at Palo Verde, both less negative compared to ED2. One notable pattern was up until a drought intensity threshold of ~40%, the climate scenarios had no effect or response in integrated-C-change at EucFACE, and the muted response from warming and eCO$_2$ Palo Verde, compared to ED2. Surprisingly, the +2K scenario switched the integrated-C-change to positive, compared to the reference case (Fig. 3g-i; red lines), potentially a physiological process in the

model to increased temperatures only that signals an anomalous resiliency response. Similar to
the results with no climate change, LPJ-GUESS remained sensitive to the intensity of drought,
with ~40% precipitation reduction being a threshold.

The models and sites differed with regard to SO and MO responses to increasing drought

severity and its interactions with warming and $eCO_2$ (related to conceptual Fig. 1d). ED2 showed
a more consistent MO response during UCEs and with additional warming and $eCO_2$ (Fig. 3;
negative integrated-C-change), especially at EucFACE, suggesting these ecosystems will remain
in a depressed carbon condition driving vegetation mortality, and/or longer recoveries. LPJ-
GUESS produced more opportunities for SO with climate change. For example, at EucFACE
$CO_2$ fertilization created small SO periods that then led to MO with increasing drought severities,
and at Palo Verde all +2K and 600 ppm led to a SO (Fig. 3j-l; Table 2).

Both models predicted that C losses due to drought interactions with increased

temperature and $eCO_2$ were less severe at the seasonally dry Palo Verde site compared to the
somewhat less seasonal, more humid EucFACE site (Table 2), which could be attributed to
higher diversity in PFT physiology at Palo Verde. Palo Verde's community composition that
emerged following drought included either three (LPJ-GUESS) or four (ED2) PFTs, while only a
single PFT existed at EucFACE. With rising temperatures under climate change, UCEs will be
hotter and drier. Nine out of the twelve simulations with both +2K and 600 ppm $CO_2$, and all but
one +2K and 800 ppm $CO_2$ produced a negative integrated-C-change, implying stronger C losses
and/or longer recovery times when droughts are exacerbated by increasing temperatures (Table

2).


**4 Discussion**

Vegetation demographic models (VDMs) allowed us to uniquely explore two hypotheses

regarding a range of modeled response of terrestrial ecosystems to unprecedented climate
extremes (UCEs), and setting the stage for the following perspectives to help guide future
research. Key model results include strong nonlinearities (Hypothesis H1) in C response to
extreme drought *intensities* in LPJ-GUESS and alternatively drought *durations* in ED2 (at one of
two sites), with differences in thresholds between the two models and ecosystems. These
nonlinearities may arise from multiple mechanisms that we begin to investigate here, including
shifts in plant hydraulics or other functional traits, C allocation, phenology, stand size-structure
and/or age demography, and compositional changes, all which vary among ecosystem types. A
critical look of driving model mechanisms, which emerged from the hypothetical drought
simulations used here, are summarized in Table 3. The models also show exacerbated biomass
loss and recovery times in the majority of our scenarios of warming and $eCO_2$, supporting
Hypothesis H2. Below, we discuss the underlying mechanisms that drive simulated ecosystem
response to UCEs using the models and sites as conceptual "experimental tools" and
observational evidence from the literature. We focus on two temporal stages of the UCE: The
pre-drought ecosystem stage characterized as the quasi-stable state of the ecosystem prior to a
UCE, which can mediate ecosystem resistance and disturbance impact, and the post-drought
recovery stage (Table 1).

**4.1 The role of ecosystem processes and states prior to UCEs**
**4.1.1 The role of phenology and phenological strategies prior to UCEs:**
Observations show that diversity of deciduousness contributes to successful alternative
strategies for tropical forest response to water stress (Williams et al., 2008). For example, during
the severe 1997 El Nino drought, brevi-deciduous trees and deciduous stem-succulents within a
tropical dry site in Guanacaste Costa Rica retained leaves during the extreme wet-season
drought, behaving differently than during normal dry seasons (Borchert et al., 2002). Both
models here predict that neither seasonal deciduousness, nor drought-deciduous phenology at the
seasonally dry tropical forest, Palo Verde (which consists of trees with different leaf
phenological strategies), act to buffer the forest from a large drop in LAI during UCEs (Fig. S1a-
b). Even with this large decrease in LAI, ED2 predicted a very weak biomass loss at the time of
UCEs (Fig. 2a), suggesting large-scale leaf loss is not a direct mechanism of plant mortality in
ED2. Leaf loss is one component of total carbon turnover flux equations in terrestrial models, in
addition to woody loss, fine-roots, and reproductive tissues. Having a better understanding of
when extreme levels of phenological turnover contribute to stand-level mortality could be
improved. Among other turnover hypothesis explored, Pugh et al. (2020) found that phenological
turnover fluxes where just as important as mortality fluxes in driving forest turnover time in the
VDMs: LPJ-GUESS, CABLE-POP, ORCHIDEE, but not the LSM JULES. At the EucFACE
site prior to the simulated extreme drought, LPJ-GUESS displayed strong inter-annual variability
in LAI (Fig. S1a-b). This capability of large swings in LAI (5.8 to 0.8) by LPJ-GUESS could
contribute to model uncertainty and the considerable mortality response at EucFACE. Modeled
LAI was the largest source of variability in another ecosystem model, CABLE, when evaluating
the simulated response to $CO_2$ fertilization (Li et al., 2018). VDMs could be improved by better
capturing different plant phenological responses to UCEs by better representing a range of leaf-
level morphological and physiological characteristics relevant to plant-water relations such as
leaf age, retention of young leaves even during extreme droughts, (Borchert et al., (2002)), and
variation in hydraulic traits as a function of leaf habit (Vargas et al., (2021)) (Table 3). Two such
examples are seen in the FATES model where the possibility for "trimming" the lowest leaf
layer can occur when leaves are in negative carbon balance due to light limitation thus
optimizing maintenance costs and carbon gain, as well as leaf age classifications providing
variations in leaf productivity and turnover.

**4.1.2 The role of plant hydraulics prior to UCEs:**
Susceptibility of plants to hydraulic stress is one of the strongest determinants of
vulnerability to drought, with loss of hydraulic conductivity being a major predictor of drought
mortality in temperate (McDowell et al., 2013; Anderegg et al., 2015; Sperry and Love, 2015;
Venturas et al., 2021) and tropical forests (Rowland et al., 2015; Adams et al., 2017b), as well as
a tractable mortality mechanism to represent in process-based models (Choat et al., 2018,
Kennedy et al., 2019). Both LPJ-GUESS and ED2 exhibited a wide range in amount and pattern
of plant-available-water prior to drought (Fig. S1c-d), contributing to large differences in UCE
response. LPJ-GUESS, which does not simulate hydrodynamics, predicted lower total plant-
available-water at both sites compared to ED2, and subsequently simulated greater mortality and
a greater increase in plant-available-water right after the UCEs as a result of less water demand.
Due to ED2 using a static mortality threshold from conductivity loss (88%), it likely does not
accurately reproduce the wide range of observations of drought-induced mortality. In ED2, large
trees, with longer distances to transport water, were at higher risk and suffered higher mortality
(Fig. 4), demonstrating how stand demography, size structure, and tapering of xylem conduits
can play an important role in ecosystem models (Petit et al., 2008; Fisher et al., 2018). Of the
VDMs that are beginning to incorporate a continuum of hydrodynamics (e.g., ED2 (described in
Methods 2.1 section) and FATES-HYDRO (Fang et al., 2022, based on Christoffersen et al.,
2016), they are able to solve for transient water from soils to roots, through the plant and connect
with transpiration demands. Therefore instead of the plant water stress function being based on
soil water potentials, it is replaced with more realistic connections with leaf water potentials.
Mortality is then caused by hydraulic failure via embolism controlled by the critical water
potential ($P_{50}$) that leads to 50% loss of hydraulic conductivity. For advancements in tree level
hydrodynamic modeling see the FETCH3 model (Silva et al., 2022), for justification for plant
hydrodynamics in conjunction with multi-layer vertical canopy profiles see Bonan et al., (2021).
There are strong interdependencies and related mechanisms connecting both hydraulic failure
(e.g., low soil moisture availability) and C limitation (e.g., stomatal closure) during drought
(McDowell et al., 2008; Adams et al., 2017b), and these interactions should be incorporated in
ecosystem modeling and further explored (Table 3).
**4.1.3. The role of carbon allocation prior to UCEs:**

Plants have a variety of strategies to buffer vulnerability to water and nutrient stress

caused by extreme droughts, such as allocating more C to deep roots (Joslin et al., 2000; Schenk
and Jackson, 2005), investing in mycorrhizal fungi (Rapparini and Peñuelas, 2014), or reducing
leaf area without shifting leaf nutrient content (Pilon et al., 1996). Alternatively, presence of
deep roots doesn't necessarily lead to deep soil moisture utilization, as seen in a 6-year
Amazonian throughfall exclusion experiment where deep root water uptake was still limited,
even with high volumetric water content (Markewitz et al., 2010). Elevated $CO_2$ alone will
enhance growth and water-use efficiency (Keenan et al., 2013), reducing susceptibility to
drought. However, such increased productivity within a forest stand, and associated structural
overshoot during favorable climate windows, can also be reversed by increased competition for
light, nutrients, and water during unfavorable UCEs – potentially leading to mortality overshoot
(Fig. 1d) and higher C loss. Mortality overshoot, as a result of structural overshoot, could be an
explanation for the negative integrated-C-change (i.e., C loss) in the majority of $eCO_2$-only
simulations (18 out of 24 scenarios; Table 2).

Effects of $CO_2$ fertilization on plant C allocation strategies are uncertain. As a result,

ecosystem models differ in their assumptions on controls of C allocation in response to $eCO_2$,
leading to divergent plant C use efficiencies (Fleischer et al., 2019). Global scale terrestrial
models are beginning to include optimal dynamic C allocation schemes, over fixed ratios, that
account for concurrent environmental constraints on plants, such as water, and adjust allocation
based on resource availability such as in LM3-PPA (Weng et al., 2015), but the representation of
C allocation is still debated and progressing (De Kauwe et al., 2014; Montané et al., 2017; Reyes
et al., 2017). Options for carbon allocation strategies can based on the allometric partitioning
theory (i.e., allocation follows a power allometry function between plant size and organs which
is insensitive to environmental conditions; Niklas, 1993), as an alternative to ratio-based optimal
partitioning theory (i.e., allocation to plant organs based on the most limiting resources)
(McCarthy and Enquist, 2007) or fixed ratios (Table 3), and the strategies should be further
investigated particularly due to VDMs substantial use of allometric relationships. A meta-
analysis of 164 studies found that allometric partitioning theory outperformed optimal
partitioning theory in explaining drought-induced changes in C allocation (Eziz et al., 2017).
Further eco-evolutionarily-based approaches such as optimal response or game-theoretic
optimization, as well as entropy-based approaches are useful when wanting to simulate higher
levels of complexity (reviewed in Franklin et al. 2012). With more frequent UCEs and the need
for plants to reduce water consumption, a shift in the optimal strategy of allocation between
leaves and fine roots should change. The goal functions (e.g., fitness proxy) used in optimal
response modeling can account for these shifts in costs and benefits of allocation between all
organs (Franklin et al. 2009, 2012).

**4.1.4 The role of plant carbon storage prior to UCEs:**

Studies of neotropical and temperate seedlings show that pre-drought storage of non-

structural carbohydrates (NSCs) provides the resources needed for growth, respiration
osmoregulation, and phloem transport when stomata close during subsequent periods of water
stress (Myers and Kitajima, 2007; Dietze and Matthes, 2014; O'Brien et al*.,* 2014). Furthermore,
direct correlations have been shown between NSC depletion and embolism accumulation, and
the degree of pre-stress reserves and utilization of soluble sugars (Tomasella et al., 2020). The
amount of NSC storage required to mitigate plant mortality during C starvation and interactions
with hydraulic failure from severe drought is difficult to quantify, due to the many roles of NSCs
in plant function and metabolism (Dietze and Matthes, 2014). For example, NSCs were not
depleted after 13 years of experimental drought in the Brazilian Amazon (Rowland et al., 2015).
As atmospheric $CO_2$ increases with climate change, NSC concentrations may increase, as seen in
manipulation experiments (Coley, 2002), but interactions with heat, water stress, enhanced leaf
shedding, and nutrient limitation complicates this relationship, and needs to be further explored.
Despite the recognition of the critical role that plant hydraulic functioning and NSCs play in tree
resilience to extremes, knowledge gaps and uncertainties preclude fully incorporating these
processes into ecosystem models.

Compared to ED2, LPJ-GUESS predicted low plant carbon storage (a model proxy for

NSCs) prior to and during drought, and at times became negative, thereby creating C costs (Fig.
S2a-b), leading to C starvation and potentially explaining the larger biomass loss in LPJ-GUESS
at both sites. Alternatively, ED2 maintained higher levels of NSCs providing a buffer to stress,
and mitigating the negative effects of drought. Maintenance of NSCs in ED2, even during
prolonged drought (at EucFACE) is due to: (1) trees resorbing a fraction of leaf C during leaf
shedding, (2) no maintenance costs for NSC storage in the current version, and (3) no allocation
of NSCs to structural growth until NSC storage surpasses a threshold (the amount of C needed to
build a full canopy of leaves and associated fine roots), allowing for a buffer to accumulate. In
LPJ-GUESS, accumulation and depletion of NSC is recorded as a 'C debt' being paid back in
later years. The contrasting responses of the two models to drought, and the likely role of NSCs
in explaining differences in model behavior, highlights the need to better understand NSC
dynamics and to accurately represent the relevant processes in models (Richardson et al., 2013;
Dietze and Matthes, 2014). More observations of C accumulation patterns and how/where NSCs
drive growth, respiration, transport and cellular water relations would enable a more realistic
implementation of NSC dynamics in models (Table 3).

**4.1.5 Role of functional trait diversity prior to UCEs:**

Currently LPJ-GUESS simulates the Palo Verde community using three PFTs, while ED2 uses

four PFTs that differ in photosynthetic and hydraulic traits. The community composition simulated by
ED2 is shown to be more resistant to UCEs compared to LPJ-GUESS (Fig. 5), perhaps due to
relatively higher functional diversity (via more PFTs with additional phenological and hydraulic
diversity). This additional diversity helps to buffer ecosystem response to drought by allowing more
tolerant PFTs to benefit from reductions in less-tolerant PFTs, thus buffering reductions in ecosystem
function (Anderegg et al., 2018). Higher diversity ecosystems were found to protect individual species
from negative effects of drought (Aguirre et al., 2021) and enhance productivity resilience following
wildfire (Spasojevic et al., 2016); thus, functionally diverse communities may be key to enhancing
tolerance to rising environmental stress.
Recent efforts to consolidate information on plant traits (Reich et al., 2007; Kattge et al., 2011)
have contributed to identifying relationships that can impact community-level drought responses
(Skelton et al., 2015; Anderegg et al., 2016a; Uriarte et al., 2016; Greenwood et al., 2017), such as
life-history characteristics, and strategies of resource acquisition and conservation as predictors of
ecosystem resistance (MacGillivray et al., 1995; Ruppert et al., 2015). While adding plant trait
complexity in ESMs may be required to accurately simulate key vegetation dynamics, it necessitates
more detailed parameterizations of processes that are not explicitly resolved (Luo et al., 2012). Further
investigation of how VDMs represent interactions leading to functional diversity shifts is crucial to
this issue. Enquist and Enquist, (2011), as an example, show that long-term patterns of drought (20-
years) have led to increases in drought-tolerant dry forest species, which could modulate resistance to
future droughts. Higher diversity of plant physiological traits and drought-resistance strategies is
expected to enhance community resistance to drought, and models should account for shifts in diverse
functionality (Table 3).

**4.2 The role of ecosystem processes and states in post-UCE recovery**
**4.2.1 The role of soil water resources post-UCEs:**
Our simulation results generally demonstrated a fast recovery of plant-available-water
and LAI at both sites (Fig. S1). Annual plant-available-water substantially increased right after
drought by an average of 163 mm at Palo Verde and 213 mm at EucFACE in the LPJ-GUESS
simulations, compared to much lower increases in ED2 (50 mm and 12 mm at Palo Verde and
EucFACE). This increase in available water post-drought can be attributed to reduced stand
density and water competition (Fig. S2c-d; diamonds vs. circles), alleviating the demand for soil
resources (water) and subsequent stress, which has also been shown in observations (McDowell
et al., 2006; D'Amato et al., 2013). After large canopy tree mortality events there can be
relatively rapid recovery of forest biogeochemical and hydrological fluxes (Biederman et al.,
2015; Anderegg et al., 2016b; Biederman et al., 2016). These crucial fluxes strongly influence
plant regeneration and regrowth, which can buffer ecosystem vulnerability to future extreme
droughts. However, this enhanced productivity has a limit. In a scenario where UCEs continue to
intensify, causing greater reductions in soil water and reduced ecosystem recovery potential, the
SO growth that typically occurs after UCEs may be dampened (Fig. 1d). In water-limited
locations, similar to the dry forest sites used here, initial forest recovery from droughts were
faster due to thinning induced competitive-release of the surviving trees, and shallow roots not
having to compete with neighboring trees for water, allowing for more effective water user
(Tague and Moritz, 2019), stressing the importance of root competition and distribution in
models (Goulden and Bales, 2019). Tague and Moritz, (2019) also reported that this increased
water use efficiency and SO ultimately lead to water stress and related declines in productivity,
similar to the MO concept (Jump et al., 2017; McDowell et al., 2006). Since a core strength of
VDMs is predicting stand demography during recovery, improved quantification of density-
dependent competition following stand dieback would be beneficial for model benchmarking
(Table 3).

**4.2.2 The role of lagged turnover and secondary stressors post-UCEs:**

Time lags in forest compositional response and survival to drought could indicate

community resistance or shifts to more competitive species and competitive exclusion. During a
15-year recovery period from extreme drought at Palo Verde, LPJ-GUESS predicted an increase
in stem density (stems $m^2$ $yr^{-1}$) (Fig. S2c) compared to ED2, which predicted almost no impact in
stem recovery. The mortality "spike" in ED2 due to drought was muted and slightly delayed,
contributing to ED2's lower biomass loss and more stable behavior of plant processes over time
at Palo Verde. At EucFACE, both models exhibited a pronounced lag effect in stem turnover
response, i.e. ~8-12 years after drought (Fig. S2d). After about a decade, strong recoveries and
increased stem density occurred, which in ED2 was followed by delayed mortality/thinning of
stems. Delayed tree mortality after droughts is common due to optimizing carbon allocation and
growth (Trugman et al., 2018), but typically only up to several years post-drought, not a decade
or more as seen in the model.

The versions of the VDMs used here do not directly consider post-drought secondary

stressors such as infestation by insects or pathogens, and the subsequent repair costs due to stress
damage, which could substantially slow the recovery of surviving trees. Forest ecologists have
long recognized the susceptibility of trees under stress, particularly drought, to insect attacks and

| 697 | pathogens (Anderegg et al., 2015). Tight connections between drought conditions and increased |
| 698 | mountain pine beetle activity have been observed (Chapman et al., 2012; Creeden et al., 2014), |
| 699 | and can ultimately lead to increased tree mortality (Hubbard et al., 2013). Leaf defoliation is a |
| 700 | major concern from insect outbreaks following droughts, and can have large impacts on C |
| 701 | cycling, plant productivity, and C sequestration (Amiro et al., 2010; Clark et al., 2010; Medvigy |
| 702 | et al., 2012). Implementing these secondary stressors in models could slow the rate of post-UCE |
| 703 | recovery and lead to increased post-UCEs tree mortality. |

### 4.2.3 The role of stand demography post-UCEs:

| 706 | Change in stand structure is an important model process to capture, because large trees |
| 707 | have important effects on C storage, community resource competition, and hydrology |
| 708 | (Wullschleger et al., 2001) (Table 3), and maintaining a positive carbohydrate balance is |
| 709 | beneficial in sustaining (or repairing) hydraulic viability (McDowell et al., 2011). There is |
| 710 | increasing evidence, both theoretical (McDowell and Allen, 2015) and empirical (Bennett et al., |
| 711 | 2015; Rowland et al., 2015; Stovall et al., 2019), that large trees (particularly tall trees with high |
| 712 | leaf area) contribute to the dominant fraction of dead biomass after drought events. Under rising |
| 713 | temperatures (and decreasing precipitation), VPD will increase, leading to a higher likelihood of |
| 714 | large tree death (Eamus et al., 2013; Stovall et al., 2019), driving MO events as hypothesized in |
| 715 | Fig. 1d. Consistent with this expectation, ED2 predicted that the largest trees (>100 cm) |
| 716 | experienced the largest decreases in basal area to compared to all other size classes (Fig. 4). This |
| 717 | drought-induced partial dieback and mortality of large dominant trees has substantial impacts on |
| 718 | community-level C dynamics, as long-term sequestered C is liberated during the decay of new |
| 719 | dead wood (Palace et al., 2008; Potter et al., 2011). In ED2, the intermediate size class (60 - 80 |
| 720 | cm) increased in basal area following large-tree death, taking advantage of the newly open |
| 721 | canopy space. However, small size classes do not necessarily benefit from canopy dieback. For |
| 722 | example, in a dry tropical forest, prolonged drought led to a decrease in understory species and |
| 723 | small-sized stems (Enquist and Enquist, 2011). |
| 724 | Due to VDMs being able to exhibit dynamic biogeography they are more useful at |
| 725 | predicting shifts in community composition beyond LSMs capabilities. Further areas of |
| 726 | advancement (described in Franklin et al. (2020)) is including models of natural selection, self- |
| 727 | organization, and entropy maximization which can substantially improve community dynamic |

responses in varying environments such as UCEs. Eco-evolutionary optimality (EEO) theory can
also help improve functional trait representation in global process-based models (reviewed in
Harrison et al., 2021), through hypotheses in plant trait trade-offs and mechanistic links between
processes such as resource demand, acquisition, and plant's competitiveness and survival; traits
associated with high degrees of sensitivity in models. The power of prognostic VDMs to predict
shifts in demography and community migration with climate change is large, but rarely is being
constrained with plant-level EEO theory, and thus will likely need to use stand level competition
and coexistence principles of how plants self-organize (Franklin et al. 2020).

**4.2.4 The role of functional trait diversity & plant hydraulics post-UCEs:**

In field experiments, higher disturbance rates have shifted the recovery trajectory and
competition of the plant community towards one that is composed of opportunistic, fast-growing
pioneer tree species, grasses (Shiels et al., 2010; Carreño-Rocabado et al., 2012), and/or
deciduous species, as also seen in model results (Hickler et al., 2004). In the treatments presented
here, deciduous PFT types were also the strongest to recover after 15 years in both models,
surpassing pre-drought values (Fig. 5). It should be noted that ED2 exhibited a strong recovery in
the evergreen PFT as well, inconsistent with the above literature (Fig. 5b). PFTs in ED2 respond
to drought conditions via stomatal closure and leaf shedding, buffering stem water potentials
from falling below a set mortality threshold (i.e., 88% of loss in conductivity). This conductivity
threshold may need to be reconsidered if further examination reveals an unrealistic advantage
under drought conditions for evergreen trees, which exhibited a lower impact from droughts
(compared to deciduous and brevi-deciduous PFTs) in ED2. Nitrogen cycling feedbacks were
not investigated here, but could also be an explanation for a strong evergreen PFT recovery.

Recovery of surviving trees could be hindered by the high cost of replacing damaged
xylem associated with cavitation (McDowell et al., 2008; Brodribb et al., 2010). Many studies
have identified "drought legacy" effects of delayed growth or gross primary productivity
following drought (Anderegg et al., 2015; Schwalm et al., 2017) and the magnitude of these
legacies across species correlates with the hydraulic risks taken during drought itself (Anderegg
et al., 2015). The conditions under which xylem can be refilled remain controversial, but it seems
likely that many species, particularly gymnosperms, may need to entirely replace damaged
xylem (Sperry et al., 2002), and trees worldwide operate within narrow hydraulic safety margins,
suggesting that trees in all biomes are vulnerable to drought (Choat et al., 2012). The amount of
damaged xylem from a given drought event and recovery rates also vary across trees of different
sizes (Anderegg et al., 2018).

Plasticity in nutrient acquisition traits, intraspecific variation in plant hydraulic traits

(Anderegg et al., 2015), and changes in allometry (e.g., Huber values) can have large effects on
acclimation to extreme droughts. This suggests some capacity for physiological adaptation to
extreme drought, as seen by short-term negative effects from drought and heat extremes being
compensated for in the longer term (Dreesen et al., 2014). Still, given the shift towards more
extreme droughts with climate change, vegetation mortality thresholds are likely to be exceeded,
as reported in Amazonian long-term plots where mortality of wet-affiliated genera has increased
while simultaneously new recruits of dry-affiliated genera are also increasing (Esquivel-Muelbert
et al., 2019). Increasing occurrences of heat events, water stress and high VPD will lead to
extended closure of stomata to avoid cavitation, progressively reducing $CO_2$ enrichment benefits
(Allen et al., 2015). Where $CO_2$ fertilization has been seen to partially offset the risk of
increasing temperatures, the risk response was mediated by plant hydraulic traits (Liu et al.,
2017) using a soil–plant–atmosphere continuum (SPAC) model, yet interactions with novel
extreme droughts were not considered. The VDM simulations suggest that the combination of
elevated warming and potential structural overshoot from $eCO_2$ (or inaccurate representation in
NSCs allocation/usage priority) will exacerbate consequences of UCEs by reductions in both C
stocks and post-drought biomass recovery speeds (Fig. 3). Therefore, future UCE recovery may
not be easily predicted from observations of historical post-disturbance recovery. An associated
area for further investigation is to better understand the hypothesized interplay between
amplified mortality from hotter UCEs followed by structural overshoot regrowth during wetter
periods (Fig. 1d), which could potentially lead to continual large swings in MO and SO and
vulnerable net ecosystem C fluxes through time (Table 3).

**5 Summary of perspectives for model advancement**

Model limitations and unknowns exposed by our simulations and literature review

highlight current challenges in our ability to understand and forecast UCE effects on ecosystems.
These limitations reflect a general lack of empirical experiments focused on UCEs. Insufficient
data means that relevant processes may currently be poorly represented in models, and models
may then misrepresent C losses during UCEs. The two VDMs used here had different
sensitivities to drought duration and intensity. These model uncertainties could potentially be
addressed by improved datasets on thresholds of conductivity loss at high drought intensities, the
role of trait diversity (e.g., different strategies of drought deciduousness and EEO theory) in
buffering ecosystem drought responses, and a better grasp of allocation to plant C storage stocks
before, during, and after multi-year droughts. Our study takes some initial steps to identify and
assess model gaps in terms of mechanisms and magnitudes of responses to UCEs, which can
then be used to inform and develop field experiments targeting key knowledge gaps as well as to
prioritize ongoing model development (Table 3). Our intention was not to do an exhaustive list
of UCE simulation experiments, and additional modeling perturbations and experiments would
be useful outcomes of future studies. For example, we begin to investigate duration of droughts
but we did not consider frequency of back-to-back UCEs. This iterative model-experiment
framework of using VDMs as hypothesis testing tools offers strong potential to drive progress in
improving our understanding of terrestrial ecosystem responses to UCEs and climate feedbacks,
while informing the development of the next generation of models.
*Code Availability.* The source code for the ED2 model can be downloaded and available publicly
at https://github.com/EDmodel/ED2. The source code for the LPJ-GUESS model can be
downloaded and available publicly at http://web.nateko.lu.se/lpj-guess/download.html. All model
simulation data will be available in a Dryad repository.

*Data Availability.* Authors received the required permissions to use the site level meteorological
data used in this study. Otherwise, no ecological or biological data were used in this study.

*Author Contributions*. JH wrote the manuscript with significant contributions from AR, BS, JD,
DM, with input and contributions from all authors. XX and MM were the primary leads running
the model simulations, with model assistance and strong feedback from DM and BS. All authors
made contributions to this article, and agree to submission.

*Competing Interests.* The contact author has declared that neither they nor their co-authors have
any competing interests.

*Special Issue Statement.* Special Issue titled "Ecosystem experiments as a window to future
carbon, water, and nutrient cycling in terrestrial ecosystems"

*Financial Support*: Funding for the meetings that facilitated this work was provided by NSF-
DEB-0955771: An Integrated Network for Terrestrial Ecosystem Research on Feedbacks to the
Atmosphere and ClimatE (INTERFACE): Linking experimentalists, ecosystem modelers, and
Earth System modelers, hosted by Purdue University; as well as Climate Change Manipulation
Experiments in Terrestrial Ecosystems: Networking and Outreach (COST action ClimMani –
ES1308), led by the University of Copenhagen. J.A. Holm's time was supported as part of the
Next Generation Ecosystem Experiments-Tropics, funded by the U.S. Department of Energy,
Office of Science, Office of Biological and Environmental Research under Contract DE-AC02-
05CH11231. AR acknowledges funding from CLIMAX Project funded by Belmont Forum and
the German Federal Ministry of Education and Research (BMBF). BS and MM acknowledge
support from the Strategic Research Area MERGE. W.R.L.A. acknowledges funding from the
University of Utah Global Change and Sustainability Center, NSF Grant 1714972, and the
USDA National Institute of Food and Agriculture, Agricultural and Food Research Initiative
Competitive Programme, Ecosystem Services and Agro-ecosystem Management, grant no. 2018-
67019-27850. JL acknowledges support from the Northern Research Station of the USDA Forest
Service (agreement 16-JV-11242306-050) and a sabbatical fellowship from sDiv, the Synthesis
Centre of iDiv (DFG FZT 118, 202548816). CDA acknowledges support from the USGS Land
Change Science R&D Program.

*Acknowledgements.* We thank Belinda Medlyn and David Ellsworth of the Hawkesbury Institute
for the Environment, Western Sydney University, for providing the meteorological forcing data
series for the EucFACE site, a facility supported by the Australian Government through the
Education Investment Fund and the Department of Industry and Science, in partnership with
Western Sydney University.

**Table 1.** Hypothesized plant processes and ecosystem state variables affecting pre-drought resistance and post-drought recovery in the context of unprecedented climate extremes (UCEs). The "Included in Model?" column indicates which processes or state variables are represented in each of the two models studied in this paper. The mechanisms listed in the two right columns refer to real-world ecosystems and are not necessarily represented in the ED2 and LPJ-GUESS models. Contents of the table are based on a non-exhaustive literature review, expert knowledge, and modeling results presented here. Symbols refer to the following literature sources: * Borchert et al., 2002; Williams et al., (2008); ** Dietze and Matthes, (2014); O'Brien et al., 2014; *** ENQUIST and ENQUIST, (2011); Greenwood et al., (2017); Powell et al., (2018); ^ Rowland et al., (2015); McDowell et al., (2013); Anderegg et al., (2015); ^^ Joslin et al., 2000; Markewitz et al., (2010); ^^^ Powell et al., (2018); ^^^^ Bennett et al., (2015); Rowland et al., (2015); ~ Hubbard et al., (2013); ~ ~ McDowell et al., (2006); D'Amato et al., (2013); + Zhu et al., (2018); Vargas et al., (2021); % Trugman et al., (2019); %% Franklin et al., (2012); %% Franklin et al., (2020).

| Process or State Variable | Included in model? | Mechanisms affecting pre-UCE drought resistance influencing impact | Mechanisms affecting post-UCE drought recovery |
|---|---|---|---|
| Processes | | | |
| 1) Phenology Schemes | ED2: Yes LPJ-G: Yes | - Leaf area and metabolic activity modulates vulnerability to death<br>- Drought-deciduousness reduces vulnerability to drought *, with higher water potential at turgor loss point and less leaf vulnerability to embolism [+] | - Leaf lifespan tends to increase from pioneer to late-successional species in some ecosystems (e.g., tropical forests) and is a balance between C gain and its cost |
| 2) Plant Hydraulics | ED2: Yes LPJ-G: No | - Cavitation resistance traits [^]<br>- Turgor loss, hydraulic failure (stem embolism) lead to increased plant mortality and enhanced vulnerability to secondary stressors. | - Replacement cost of damaged xylem slows recovery of surviving trees |
| 3) Dynamic Carbon Allocation | ED2: Yes LPJ-G: Yes | - Increased root allocation could offset soil water deficit under gradual onset of drought [^^]<br>- Leaf C allocation strategies should be connected to hydraulic processes [%] | - Allocation among fine roots, xylem, & leaves affects recovery time & GPP/LAI trajectory<br>- Eco-evolutionary optimality theory [%%] |

| | | | |
|---|---|---|---|
| 4) Non-Structural Carbohydrate (NSC) Storage | ED2: Yes LPJ-G: Yes | - NSCs buffer C starvation mortality due to reduced primary productivity.<br>- Maintenance of hydraulic function & avoiding hydraulic failure ** | - Low NSC could increase vulnerability to secondary stressors during recovery |
| State Variables | | | |
| 1) Plant-Soil Water Availability | ED2: Yes LPJ-G: Partly | - Low soil water potential increases risk of tree C starvation, turgor loss and hydraulic failure | - After stand dieback reduced demand for soil resources &/or reduced shading<br>- Increased soil water enhances regeneration/ regrowth, buffers vulnerability to long-term drought ~ ~ |
| 2) Plant Functional Diversity | ED2: Yes LPJ-G: Yes | - Presence of drought-tolerant species modulates resistance at community level.<br>- Shallow-rooting species more vulnerable ^^^ *** | - Changed resource spectra shift competitive balance in favor of grasses and pioneer trees |
| 3) Stand Demography | ED2: Yes LPJ-G: Yes | - Larger tree size enhances vulnerability to drought and secondary stressors due to higher maintenance costs ^^^^ | - Mortality of canopy individuals favors understory species and smaller size-classes<br>- Self-organizing principles %%% |
| 4) Compounding Stressors | ED2: No LPJ-G: No | - Reduced resistance to insects and pathogens due to physiological/mechanical/ hydraulic damage & depletion of NSC | - Infestation by insects and pathogens, repair of damage due to secondary stressors, slows recovery of surviving trees ~ |


**Table 2** Impact of eCO$_2$ and/or temperature on the integrated-C-change (kg C m$^{-2}$ yr) relative to
drought treatments with no additional warming or eCO$_2$, for both models, and both sites seen in
Fig. 3. Quantified as average and minimum integrated-C-change across all 20 drought intensities
for step-change scenarios of warming and eCO$_2$. The percentage of each scenario that was
negative in integrated-C-change (i.e., decreases in C loss). Green values represent positive
integrated-C-change.

| | | ED2 | | | LPJ-GUESS | | |
|---|---|---|---|---|---|---|---|
| **EucFACE** | | Average integrated C change | Largest integrated C change | % climate scenario was negative | Average integrated C change | Largest integrated C change | % climate scenario was negative |
| 1 year | 600 ppm | 2.2 | 0.0 | 33.3 | -74.6 | -396.6 | 36.8 |
| | 800 ppm | -10.6 | -73.0 | 50.0 | -124.1 | -416.0 | 57.9 |
| | 2K | 2.3 | -0.5 | 16.7 | 21.3 | -20.8 | 15.8 |
| | 2K, 600 ppm | 0.5 | -8.2 | 61.1 | -67.5 | -201.5 | 78.9 |
| | 2K, 800 ppm | 1.8 | -0.4 | 22.2 | -145.9 | -400.1 | 47.4 |
| 2 year | 600 ppm | -105.6 | -456.7 | 77.8 | -85.2 | -260.6 | 63.2 |
| | 800 ppm | -199.0 | -522.9 | 83.3 | -106.3 | -350.1 | 42.1 |
| | 2K | -10.3 | -34.7 | 77.8 | 14.2 | -35.2 | 31.6 |
| | 2K, 600 ppm | -204.9 | -666.1 | 77.8 | -47.6 | -128.8 | 84.2 |
| | 2K, 800 ppm | -12.4 | -61.6 | 50.0 | -167.0 | -421.9 | 68.4 |
| 4 year | 600 ppm | -125.5 | -306.2 | 83.3 | -122.6 | -277.4 | 94.7 |
| | 800 ppm | -277.1 | -423.3 | 100.0 | -212.2 | -523.7 | 89.5 |
| | 2K | -61.8 | -188.6 | 72.2 | 12.9 | -13.8 | 31.6 |
| | 2K, 600 ppm | -385.9 | -674.2 | 94.4 | -79.1 | -197.3 | 94.7 |
| | 2K, 800 ppm | -277.9 | -737.7 | 72.2 | -247.0 | -503.8 | 100.0 |
| *Average* | | -111.0 | -277.0 | 64.8 | -95.4 | -276.5 | 62.5 |
| *Palo Verde* | | ED2 | | | LPJ-GUESS | | |
| 1 year | 600 ppm | -1.6 | -6.2 | 77.8 | -11.0 | -32.4 | 78.9 |
| | 800 ppm | 6.7 | -0.2 | 11.1 | -39.2 | -154.0 | 100.0 |
| | 2K | -1.0 | -15.3 | 38.9 | -33.4 | -75.1 | 100.0 |
| | 2K, 600 ppm | 2.5 | -1.1 | 22.2 | 6.5 | -4.6 | 52.6 |
| | 2K, 800 ppm | -6.6 | -16.6 | 77.8 | -121.1 | -237.7 | 100.0 |
| 2 year | 600 ppm | 15.1 | -16.7 | 38.9 | 27.3 | -6.0 | 10.5 |
| | 800 ppm | -229.2 | -756.6 | 66.7 | 20.6 | -17.2 | 26.3 |
| | 2K | -8.2 | -71.8 | 50.0 | 32.0 | -12.7 | 15.8 |
| | 2K, 600 ppm | 24.8 | -5.7 | 11.1 | 36.2 | -1.2 | 5.3 |
| | 2K, 800 ppm | -152.9 | -348.1 | 77.8 | 8.0 | -54.5 | 36.8 |
| 4 year | 600 ppm | -11.1 | -37.3 | 94.4 | 3.4 | -25.1 | 26.3 |
| | 800 ppm | -260.2 | -694.8 | 94.4 | -25.2 | -132.6 | 57.9 |
| | 2K | -39.0 | -133.8 | 66.7 | -7.7 | -45.9 | 68.4 |
| | 2K, 600 ppm | 1.0 | -16.4 | 38.9 | 6.1 | -4.1 | 31.6 |
| | 2K, 800 ppm | -148.5 | -429.3 | 83.3 | -20.0 | -75.5 | 78.9 |
| *Average* | | -53.9 | -170.0 | 56.7 | -7.8 | -58.6 | 52.6 |


**Table 3** Summary of suggested critical look of driving mechanisms (e.g., ecosystem or plant
processes and state variables) which emerged from the hypothetical drought simulations used
here to explore for future research in manipulation experiments, data collection, and model
development and testing, as related to furthering our understanding of UCE resistance and
recovery.

| | **UCE Drought Resistance & Recovery Summary** |
|---|---|
| **Processes** | **Suggestions of driving mechanisms to further explore in data and models** |
| 1) Phenology Schemes | Represent morphological and physiological traits relevant to plant-water relations; drought- deciduousness can reduce vulnerability to drought; phenology of evergreens needs more investigation. |
| 2) Plant Hydraulics | Interactions between hydraulic failure (e.g. low soil moisture availability) and C limitation (e.g. stomatal closure) during drought should be included in models. Account for turgor loss, hydraulic failure traits, costs to recover damaged xylem. |
| 3) Dynamic Carbon Allocation | C allocation based on eco-evolutionary optimality (EEO) and allometric partitioning theory in addition, or replacing ratio-based optimal partitioning theory, and fixed ratios. Explore root allocation that could offset soil water deficits. |
| 4) Non-structural Carbohydrate (NSC) Storage | Deciding best practices for NSC representation in models. Better understanding of NSC storage required to mitigate plant mortality during C starvation and interactions with avoiding hydraulic failure during severe droughts. |
| **States Variables** | |
| 1) Plant-Soil Water Availability | Better quantification of the amount and accessibility of plant-available water for surviving trees, and tradeoff between increased structural productivity but vulnerability to subsequent droughts. Future relevance, or benefit, of lower water demand due to thinning with UCEs. |
| 2) Plant Functional Diversity | Understand how higher diversity of plant physiological traits and drought-resistance strategies will enhance community resistance to drought; models still need to account for shifts in diverse functionality, including deciduousness shifts and interplay of regrowth structural overshoot followed by amplified mortality from hotter UCEs. |
| 3) Stand Demography | Large trees more vulnerable to drought; need data on changes in C stock with UCEs in high-density smaller tree stands vs. stands with larger trees. Using 'self-organization' principles for modeling stand level competition and coexistence under UCEs. |


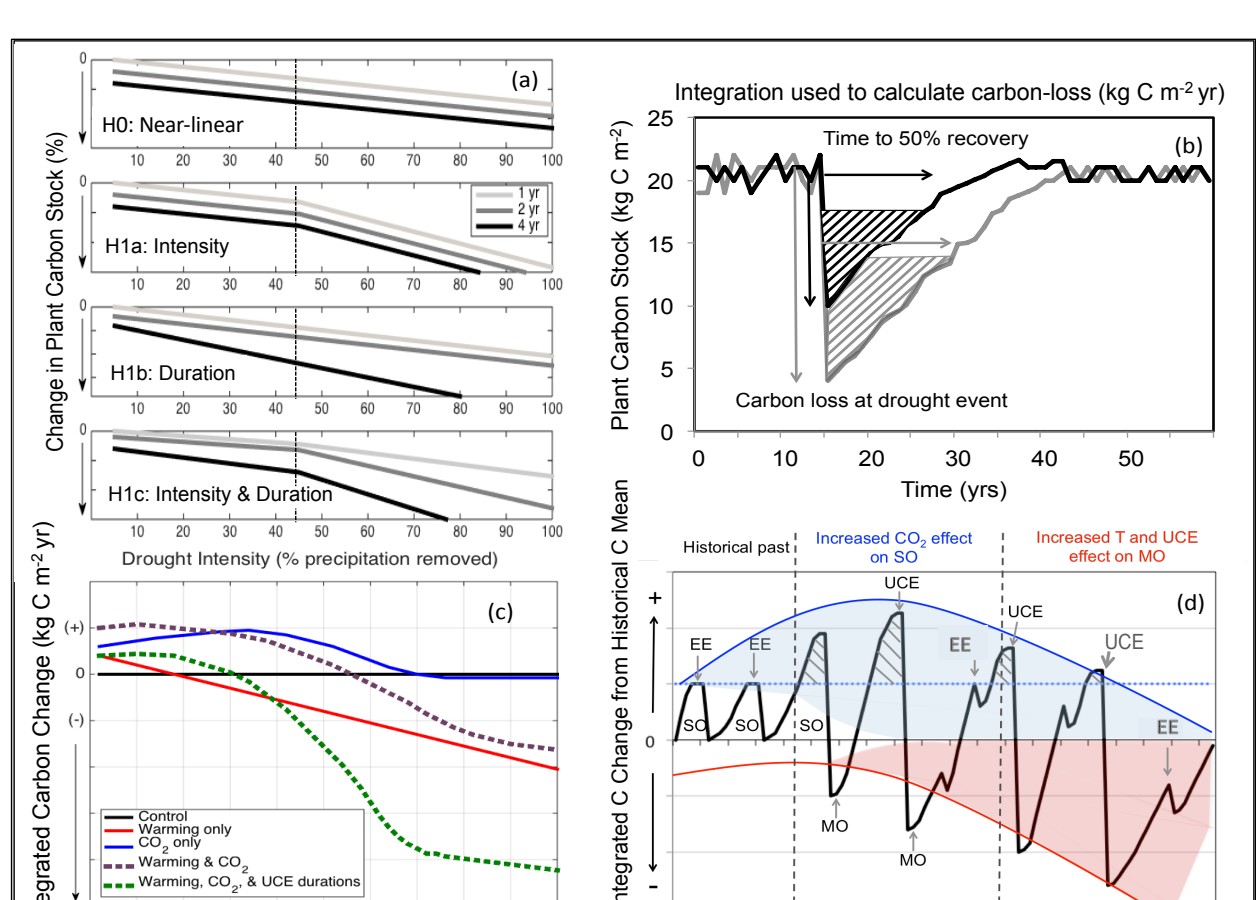


**Figure 1** Conceptual diagrams showing impacts of extreme droughts (unprecedented climate
extremes, UCEs; i.e., record-breaking droughts) on plant C stocks. (a) **Conceptual response**
**diagram**: potential loss in C stock as a function of increasing drought intensity (0-100%
precipitation removal) and drought duration (1, 2 or 4 years of drought). In this example, an
arbitrary threshold of 45% precipitation reduction and 4-year drought duration is assumed to
correspond to a UCE. The "null hypothesis" (H0, top panel) is a near-linear response of C stocks
to droughts. Alternative hypotheses include nonlinear and threshold responses to drought
intensity (H1a), drought duration via different slope responses (H1b), and combined effects of
both drought intensity and durations (H1c). (b) **Conceptualized UCE C loss diagram**:
responses of forest C stocks to a large (grey) and small (black) UCE. "Integrated-C-loss" (kg C
$m^{-2}$ yr) denotes the integral of the C loss over time and is calculated from the two arrows: the
total loss in C (kg C $m^{-2}$) due to drought, and the time (yr) to recover 50% of the pre-drought C
stock. (c) **Conceptualized UCE-climate C change diagram:** hypothetical response in terrestrial
"integrated-C-change" (kg C m$^{-2}$ yr) due to $eCO_2$ (blue line), rising temperature (red line),
interaction between $eCO_2$ and temperature (dashed purple), and combined interactions among
$eCO_2$, temperature, and UCEs of prolonged durations (green line), all relative to a reference
drought of normal duration with no warming (black line). Integrated-C-change denotes the
difference in integrated-C-loss (see panel b) between a scenario of changing climatic drivers and
the reference drought (control). (d) **Conceptual UCE amplification diagram**: hypothetical
amplified change in forest C stocks to $eCO_2$ and temperature relative to the pre-warming
historical past (based on Jump et al. (2017)). Change in C stock greater than zero indicates a
'structural overshoot' (SO) due to favorable environmental conditions and/or recovery from an
extreme drought-heat event (EE). Hashed black areas indicate a structural overshoot due to
$eCO_2$, which occurs over the historical $CO_2$ levels (dashed blue line). Initially, an $eCO_2$ effect
leads to a larger increase in structural overshoot (due to $CO_2$ fertilization), driving more extreme
vegetation mortality ('mortality overshoot' - MO) relative to historical dieback events and thus a
greater decrease in C stock. Increased warming through time increasingly counteracts any $CO_2$
fertilization effect; while the amplitude of post-UCE C stock recoveries remains large, net C
stock values eventually decline (downward curvature) due to more pronounced loss in C stocks
(and greater ecosystem state change) from hotter UCEs.
SO = structural overshoot, MO = mortality overshoot, EE = historically extreme drought-heat
event, UCE = unprecedented climate extreme.

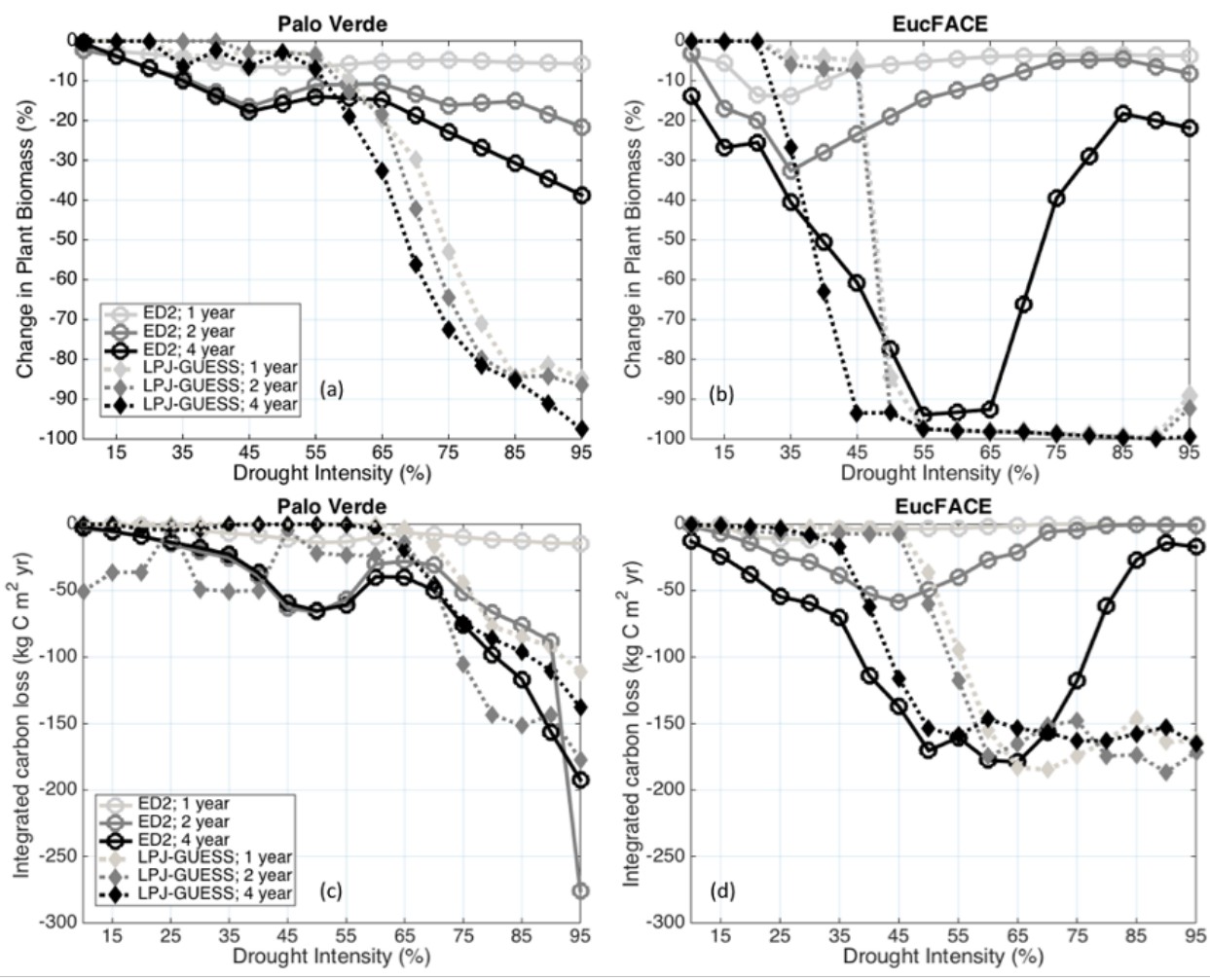


**Figure 2** Modeled change in biomass (%) at the end of drought periods of different lengths (1, 2, and 4-year droughts) and intensities (up to 95% precipitation removed) at (a) Palo Verde, and (b) EucFACE, for the ED2 and LPJ-GUESS models. Modeled integrated-C-loss (C reduction due to extreme drought integrated over time until biomass recovers to 50% of the non-drought baseline biomass) at (c) Palo Verde and (d) EucFACE.

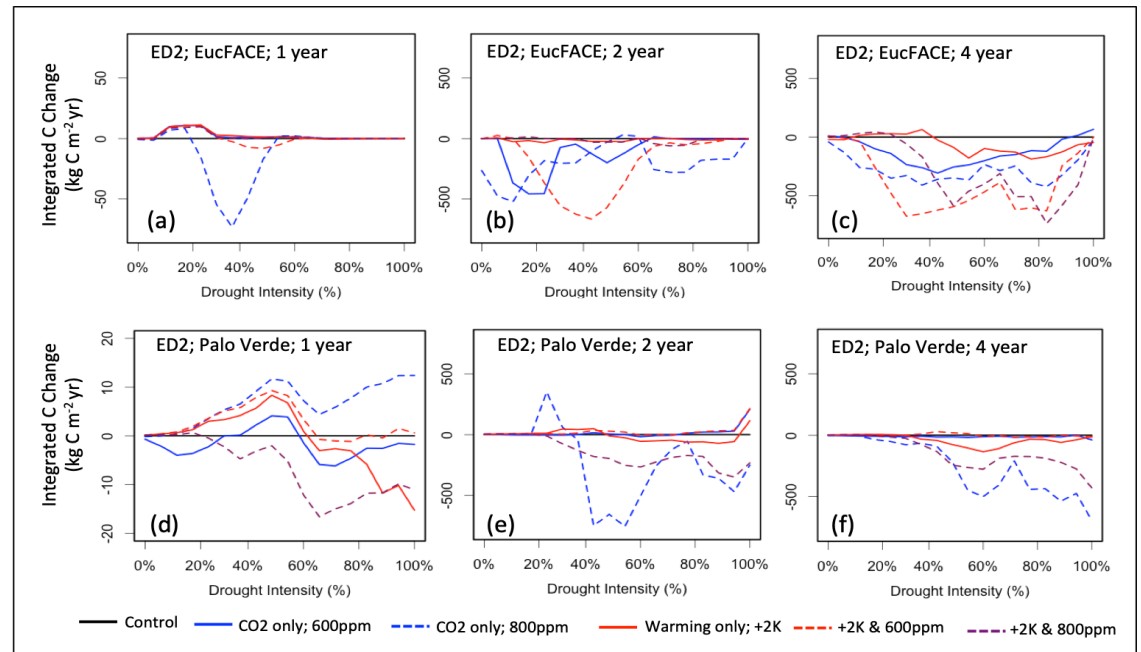

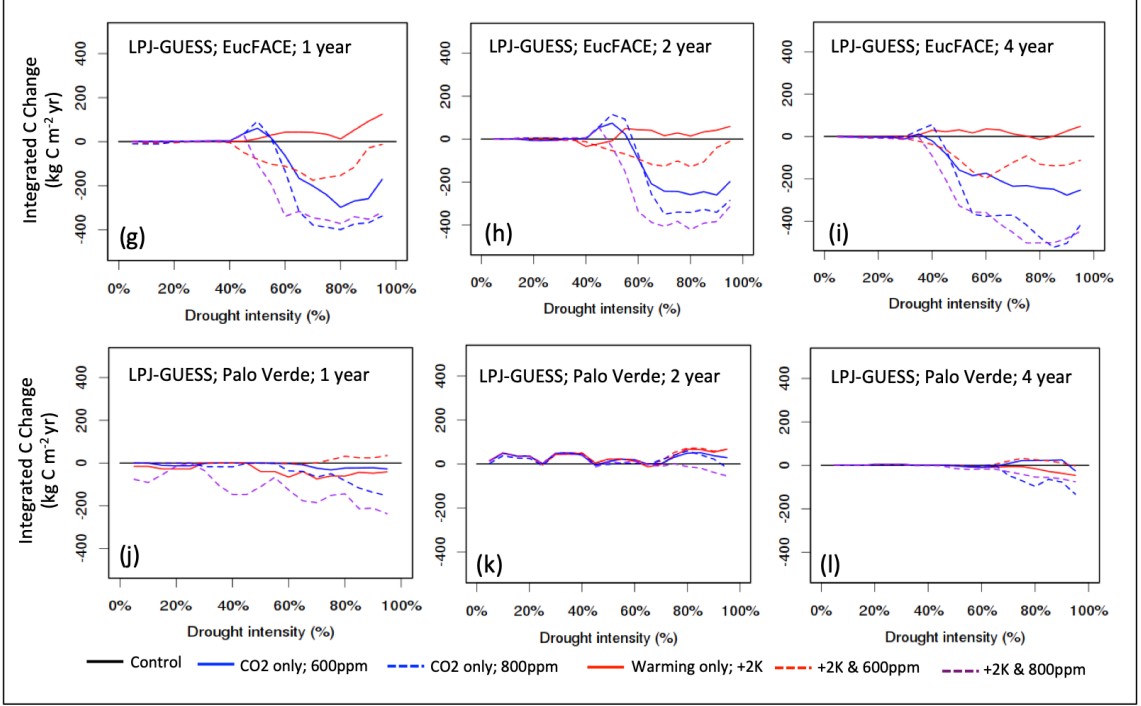

**Figure 3** Vegetation C response to interactions between drought intensity (0% to 100%
precipitation reduction), drought durations (1, 2, 4-year droughts), and idealized scenarios of
warming and eCO$_2$ compared to the reference simulation, simulated by two VDMs; ED2 (a-f)
and LPJ-GUESS (g-l) at two sites (EucFACE and Palo Verde). The scenarios include a control
(current temperature; 400 ppm atmospheric CO$_2$), two eCO$_2$ scenarios (600 ppm or 800 ppm),

elevated temperature (2 K above current), and a combination of $eCO_2$ (600 ppm or 800 ppm) and
higher temperature. Vegetation response is quantified as "integrated-C-change" (in kg C $m^{-2}$ yr;
Eq. 4), which is defined as the difference in integrated-C-losses due to drought between a given
scenario of change in climatic drivers and the control. Negative values for integrated-C-change
indicate that warming and/or $eCO_2$ leads to stronger C losses and/or longer recovery, while
positive values for integrated-C-change indicates a buffering effect.


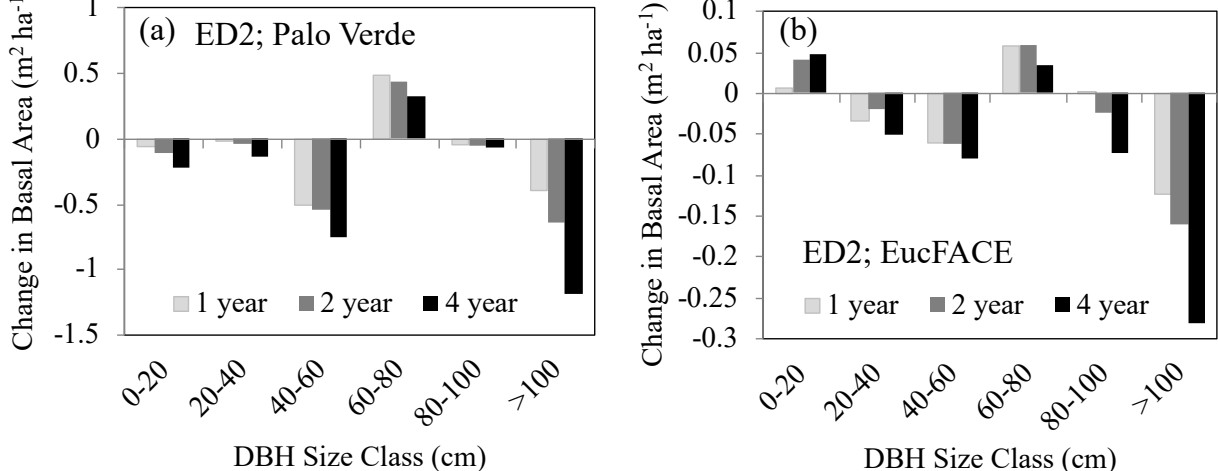


**Figure 4** Change in basal area ($m^2$ $ha^{-1}$) immediately following either 1, 2, or 4 year droughts for
six increasing size class bins (DBH, cm) as predicted by the ED2 model for (a) the Palo Verde
site, with 90% precipitation removed, and (b) the EucFACE site with 50% precipitation
removed.

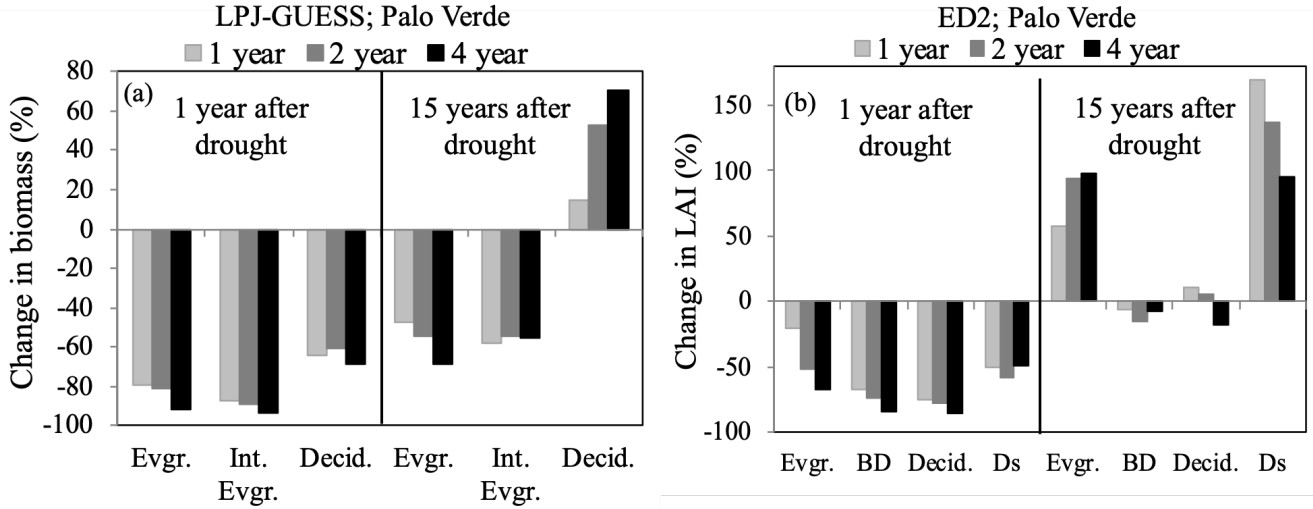

**Figure 5** Percent change in community composition, represented by plant functional type (PFT), the year following three drought durations of UCEs (1, 2, and 4-year droughts and 90% precipitation removed) as well as 15 years after droughts, for the tropical Palo Verde site by (a) LPJ-GUESS reported in biomass change, and (b) ED2 reported in LAI change. Even though Ds had the strongest recovery, it should be noted it was the least abundant PFT at this site. Evgr. = evergreen, Int. Ever. = intermediate evergreen, Decid. = deciduous, BD = brevi-deciduous, Ds = deciduous stem-succulent. EucFACE data not shown because only one PFT present (evergreen tree).

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
