# Peer review of "Exploring the impacts of unprecedented climate extremes on forest ecosystems: hypotheses to guide modeling and experimental studies"

_Biogeosciences, 2022_

## Author Response (AR1)

Sept. 16th, 2022
Dear Biogeosciences Editor and Reviewers,

We appreciate your review and comments on "Exploring the impacts of unprecedented climate extremes on forest ecosystems: hypotheses to guide modeling and experimental studies". Your time spent on this peer-review process is appreciated. Below is a point-by-point response to the reviews, and a description of how the manuscript has been revised accordingly.

Both reviewers had concerns related to comparing model results with site measurements. Such an approach would be logical for a more classic model study, such as testing model performance and subsequently using the model for temporal forecasting. However, the intention of our study was not to conduct a classic model experiment but rather a review of how models currently capture unprecedented climate extremes (UCEs) on forest ecosystems by using a handful of novel model results to guide that discussion. Our hypothesis testing approach is because data on UCEs, which would be needed for a classic model application are rare or non-existing, when extreme events have occurred, they are poorly monitored, or the extreme events that have been tested experimentally have been relatively moderate. Nevertheless, our hypothesis is that UCEs may in fact have the potential to not only have dramatic direct effects on forest ecosystems but also determine future growth and structure of such ecosystems. Therefore, we applied well tested models that have already been applied and validated at the sites used in this study. These models currently serve as our best description of the ecosystems, and therefore are well suited to test their responses to the UCEs as a kind of "hypotheses" in order to evaluate the results and to guide the need for future experiments and/or monitoring data. Our purpose is not to claim that the models are correct - most likely they are not simply because there have been no experiments or data to train and test them against UCEs.

We think many of the reviewers' concerns have now been addressed by clearly stating at the beginning of the manuscript these rationales for the study, and that this study was not intended to solely be an original research paper, but rather a road map and review paper. In order to generate an informed perspective on model responses to unprecedented climate extremes, we had to generate some original model results, which enabled our novel insights.

Best,
Jennifer Holm and co-authors

**R1 expressed concerns about:**

1. **R1: The novelty of the paper; and whether this paper was a review paper or a research paper.**

Answer: The novelty of our study is to exemplify unprecedented climate extremes (UCEs) and how models predict their impacts. In the revised manuscript, we improved the text about tighter linkages between concepts, hypotheses, and model outcomes. For example, we now emphasize that a goal of this paper is to demonstrate how to use these vegetation demographic models (VDMs) to help generate future hypotheses about and experimentally test UCEs. Therefore, we used the models and sites as conceptual "experimental" tools to investigate the given hypotheses.

We also revised the manuscript to state more clearly at the beginning that this is more of a review and "guidebook" manuscript, but based on original and well tested and published research to enable exploration of novel forest responses to such unprecedented climate extremes. Because of the "unprecedented-ness" of the extreme, they and the models have for good reasons not been tested. We believe this combination of original research with review of current limitations of models is a strong novelty in and of itself.

We would like to clarify that the original "research" aspects of this study were the new UCE model results, and specifically the integrated C-loss with sensitivity to different climate change treatments which has not previously been done with these models, and is a knowledge gap that was filled by this study. The discussion section and detailed tables are designed to be a "review" and to help guide future research, for example in terms of future needs for experiments and observations to help improve model descriptions in this context.

New text at the beginning of the discussion to make clear this is more of a review paper:
"Vegetation demographic models (VDMs) allowed us to uniquely explore two hypotheses regarding a range of modeled response of terrestrial ecosystems to unprecedented climate extremes (UCEs), and setting the stage for the following perspectives to help guide future research."

2. **R1: While one paper can only tackle so much, R1 thinks this study would benefit from additional simulation experiments.**

Answer: The scope of our paper was *not* to do many, in-depth, detailed model experiments, but rather to exemplify how models currently capture UCEs in extreme durations/lengths and interactions with climate change. Furthermore, there is little if any data available to perform such a test. Therefore, we do not feel that there would be a strong benefit to adding additional simulation experiments, which would be a large, costly endeavor and make the already very long manuscript even longer.

Also, and importantly, additional simulations (as suggested by R1) might not lead to the novelty and clarity that R1 seeks. We revised the manuscript to state that outstanding modeling perturbations and experiments would be useful outcomes of future studies. The initial modeling investigation here highlights how VDMs (as opposed to typical LSMs) can be used to answer hypotheses and guide future studies.

In the summary paragraph of the manuscript, we include the following text: "Our study takes some initial steps to identify and assess model uncertainties in terms of mechanisms and magnitudes of responses to UCEs, which can then be used to inform and develop field experiments targeting key knowledge gaps as well as to prioritize ongoing model development (Table 3). Our intention was not to do an exhaustive list of UCE simulation experiments, and outstanding modeling perturbations and experiments would be useful outcomes of future studies."

3. **R1: Lack of model validation at the two sites, and site-level observations to inform model parameterization.**

Answer: We agree that model comparison to some site-level observations is a worthy improvement to the paper. We now include site-level benchmarking observations such as biomass and leaf area index for basic model validation. We also mention that both models have already been run and validated at these sites in previously published papers (Xu et al., 2016; Medlyn et al., 2016), thus making model application possible with a "built in" reasonable degree of validation. These references were included in the original manuscript, but we failed to make a clear connection that they were publications that validated the models at the two sites.

In the methods section (Section 2.2) we added in some additional site-level observations to help the reader gain a better understanding of the site characteristics. For example, included new text such as: "stand basal area is 29.2 ($\pm$ 8.1) $m^2$ ha, stem density of 64 ($\pm$ 12) trees $ha^{-1}$....." and ".....leading to a strong seasonality in LAI ranging from 3 to 4.5, but can get as low as 1.2 $m^2$ $m^{-2}$ (Kalacska et al., 2005). "

In the results section we added this new text: As a basis for the treatment results presented here, we compared the baseline simulations (prior to drought or climate change treatments) of the two VDMs to observations at both sites for biomass and LAI (Table S2, Fig. S1). Both models had similar biomass compared to observations at Palo Verde (10.4 - 11.7 vs. 11.0 kgC $m^{-2}$), and at EucFACE biomass matched well in LPJ-GUESS (12.1 vs. 12.7 kgC $m^{-2}$) but was low in ED2 (5.6 kgC $m^{-2}$). Both models also had similar LAI to observations at Palo Verde (3.3 – 4.5 vs. 3.8 ($\pm$ 1.06) $m^2$ $m^{-2}$), and at EucFACE LAI matched well in ED2 (1.6 vs. 1.7 $m^2$ $m^{-2}$), but was high for LPJ-GUESS (3.2 $m^2$ $m^{-2}$). At EucFACE LAI ranged from 1.2 to 2.1 over a 28-month measurement period (Duursma et al., (2016), but LPJ-GUESS had very large fluctuations in annual LAI outside of these ranges (Fig. S1). These models are well

documented and investigated VDMs, with many studies that have looked into parameter uncertainty (see Supplemental Text A for select references that explore model/parameter sensitivity).

In the supplements we added site-level observations and model validation for both models, and both sites:

**Table S2.** Comparison of *in situ* observations and baseline model simulations from ED2 and LPJ-GUESS for the two example study sites, Palo Verde in Costa Rica (Kalacska et al., 2005; Xu et al., 2016) and EucFACE in Australia (Medyln et al., 2016; Duursma et al., 2016). Mean and ± standard deviation.

|  | Palo Verde Costa Rica | EucFACE Australia |
|---|---|---|
| Obs. Biomass (kgC m$^{-2}$) | 11.0 (5.2) | 12.7 (4.5) |
| ED2 Biomass (kgC m$^{-2}$) | 11.7 (0.3) | 5.6 (0.3) |
| LPJ-GUESS Biomass (kgC m$^{-2}$) | 10.4 (0.2) | 12.1 (0.2) |
| Obs. LAI (m$^2$ m$^{-2}$) | 3.8 (1.06) | 1.7 (0.6) |
| ED2 LAI (m$^2$ m$^{-2}$) | 3.3 (0.1) | 1.6 (0.2) |
| LPJ-GUESS LAI (m$^2$ m$^{-2}$) | 4.5 (0.1) | 3.2 (1.3) |

**4. R1: Reporting uncertainties in model parameters.**

Answer: The ED2 and LPJ-GUESS models are well documented and investigated VDMs, with many previous studies that have looked into parameter uncertainty. For example, we know that parameters related to plant hydraulics and non-structural carbohydrate storage have large uncertainties and thus included these parameters in the overview Table 1. We updated the supplemental material (Supplemental Text A: "Review of Model Parameter Uncertainty") to now reference additional papers that have done more in-depth investigations of specific model parameters with the largest uncertainties within each of the models, and list which processes or mechanisms were evaluated. We updated the Supplements to include these references: Oberpriller et al., 2022; Zaehle et al., 2005; Pappas et al., 2013; Jiang et al., 2012; Shiklomanov et al., 2020; Viskari et al., 2019 for more description of parameter uncertainty.

In Section 4 we more clearly state that large losses in carbon are likely linked to uncertainties in how we currently represent plant hydraulics (section 4.1.2), non-structural carbohydrate storage (section 4.1.4), or phenology diversity (section 4.1.1). We also revised the manuscript to emphasize that a goal of this paper is to demonstrate how to use these VDMs in order to help generate future hypotheses about UCEs.

Therefore, we used the models and sites as conceptual "experimental" tools to investigate the given hypotheses.

The aim of the paper was not to do in-depth, detailed model experiments with tuned parameters specific to each site, but rather to set up a general modeling framework so that hypotheses about unprecedented climate extremes can be investigated, and to provide understanding on how model behavior of physiological and ecological processes might be lacking in state-of-the-art ecosystem models in order to capture extremes. In the discussion we provide a review of where/how models could be improved in future studies, in which we also guide where future studies could look into better process representation and parameter uncertainty.

We were glad to hear that R1 thought the paper was a "very informative read" and that "the presented framework is practical and logical". As well as "the manuscript is in itself coherent, well-written, the structure is easy to follow with the aid of straightforward visualizations and tables."

**R2 expressed concerns about:**
1) **R2: It is not clear whether the main focus of this study is to introduce the newest model development or review what is missing in the current VDMs and suggest future directions, or even the combination of both.**

Answer: We revised the manuscript to clarify that our goal is to highlight what is missing in current VDMs, even after taking into account cutting edge model developments. In order to review what is lacking in current VDMs, we needed to describe the existing model frameworks and latest model developments (which might have been a little confusing). We clarified this in the revision.
We also revised the manuscript to state more clearly at the beginning that this is more of a review and "guidebook" manuscript, but informed by some original research that allowed us to address novel forest responses to climate extremes. We believe this combination of original research with a review of current limitations of models is a strong novelty in and of itself.

In the abstract we added this new text: "Here, we present a road map of how two dynamic vegetation demographic models (VDMs) can be used to investigate hypotheses surrounding ecosystem responses to UCEs (e.g., unprecedented droughts)."

2) **R2: Lack of model validation and comparisons with the observation/data, and validating against some historic drought events at these sites.**

Answer: We agree that model comparison to some site-level observations is a worthy improvement to the paper.  As discussed above, we now include site level benchmarking observations such as biomass and leaf area index for basic model validation. With regards to validating the models against historic drought events at these sites, as also pointed out by reviewer #2, we agree that this would be a logical and excellent way of testing the model outputs in another paper. It is not easy to track the effects of a drought event in reality when "extreme" events rarely coincide with field campaigns, and do not allow control of other potentially interacting factors. The goal of this paper was to explore how models represent ecosystem responses to *extreme* droughts, and identify how they are different from less intense droughts, as the responses can be nonlinear.

In section 2 of the manuscript the following text is included: "Since field data needed to evaluate UCE responses are, by definition, unavailable, we do not perform model-data comparisons. Rather, we use the model results and conceptual framework as a road map to explore our hypotheses and illustrate their implications for ecosystem responses under UCEs, not historical drought events."

**Table S2.** Comparison of *in situ* observations and baseline model simulations from ED2 and LPJ-GUESS for the two example study sites, Palo Verde in Costa Rica (Kalacska et al., 2005; Xu et al., 2016) and EucFACE in Australia (Medyln et al., 2016; Duursma et al., 2016). Mean and ± standard deviation.

|  | Palo Verde Costa Rica | EucFACE Australia |
|---|---|---|
| Obs. Biomass (kgC m$^{-2}$) | 11.0 (5.2) | 12.7 (4.5) |
| ED2 Biomass (kgC m$^{-2}$) | 11.7 (0.3) | 5.6 (0.3) |
| LPJ-GUESS Biomass (kgC m$^{-2}$) | 10.4 (0.2) | 12.1 (0.2) |
| Obs. LAI (m$^2$ m$^{-2}$) | 3.8 (1.06) | 1.7 (0.6) |
| ED2 LAI (m$^2$ m$^{-2}$) | 3.3 (0.1) | 1.6 (0.2) |
| LPJ-GUESS LAI (m$^2$ m$^{-2}$) | 4.5 (0.1) | 3.2 (1.3) |

3) **R2: LPJ-GUESS showed large swings in LAI at EucFACE site, and R2 is wondering if this is reasonable?**

Answer:  In the revised manuscript, we provided observational data of leaf area index (LAI) at the EucFACE site and compared against the large swings from the LPJ-GUESS model (Figure S1a,b).  We found that the large swings in the model are *not* typical of what has been measured at the Australian field site, and are a potential source of model biases/uncertainty.  Since this paper is more of a guidebook for how we can improve

models, this is exactly the type of discrepancy between model and observations we would like to uncover, and fix in future studies.

We updated the discussion to include: "This capability of large swings in LAI (5.8 to 0.8) by LPJ-GUESS could contribute to model uncertainty and the considerable mortality response at EucFACE. Modeled LAI was the largest source of variability in another ecosystem model, CABLE, when evaluating the simulated response to $CO_2$ fertilization (Li et al., 2018)."

[Figure]

4) **R2: Question about why these two specific study sites are selected? (And why no site level tuning was done).**

Answer: We choose these two forested sites in Australia and Costa Rica mainly because the models have already been run at these sites in previously published papers (Xu et. al., 2016; Medlyn et. al., 2016), thus making model application possible with a "built in" reasonable degree of validation. These sites also contain multiple measurements and information that allowed for previous papers to validate the models. These sites were also chosen because they span an interesting range of vegetation

types (a temperate-subtropical transitional forest in EucFACE and a seasonally dry tropical forest at PaloVerd), and are in warm, seasonally dry climates that are more likely to experience droughts in the future. In the revised manuscript we include in Section 2.2 a description of why the sites were chosen and reference the publications where model validation has already occurred. We also emphasize that the purpose of this paper was not to do site comparisons among many different sites, but rather to do some specific hypothesis testing at the best sites available.

New text: "To exemplify how VDMs can be tools to explore new hypotheses related to UCEs we applied the models at two field sites, that were chosen due to being extensively studied and the models used here have already been run at these sites and previously benchmarked against field data (see Xu et al., 2016; Medlyn et al., 2016; Medvigy et al., 2019 for model-data validation). The purpose of this paper was not to do a large multi-site comparison, but rather just select a few for hypothesis testing. In addition, the two sites span a range of vegetation types and are in warm, seasonally dry climates that are more likely to experience droughts in the future (Allen et al., 2017)."

"The two models were previously tuned for each site (Xu et al., 2016; Medlyn et al., 2016), and no additional site-level parameter tuning was conducted here due to evaluating responses from hypothetical UCEs."

5) **R2: Would like to see a tighter connection between the simulation results and discussion section which describes what is lacking in VDMs, by providing more detailed descriptions of the two models.**

Answer: In the revised manuscript, we improved the text about tighter linkages between concepts, hypotheses, and model outcomes. We renamed section 2.3 in the methods to be: "2.3 Linking concepts, hypotheses, and model outcomes", where we describe the concepts to explore how UCEs are modulated by climate change in the model results.

In addition, in Section 4 we more clearly state that large losses in carbon are likely linked to uncertainties in how we currently represent plant hydraulics (section 4.1.2), non-structural carbohydrate storage (section 4.1.4), or phenology diversity (section 4.1.1). We also revised the manuscript to emphasize that a goal of this paper is to demonstrate how to use these VDMs in order to help generate future hypotheses about UCEs. Therefore, we used the models and sites as conceptual "experimental" tools to investigate the given hypotheses.  In the supplemental material (Supplemental Text A: "Review of Model Parameter Uncertainty") we now reference additional papers that have done more through investigations of specific model processes within each of the models, and list which processes or mechanisms were evaluated.

We updated the Supplements to include these references: Oberpriller et al., 2022; Zaehle et al., 2005; Pappas et al., 2013; Jiang et al., 2012; Shiklomanov et al., 2020; Viskari et al., 2019 for more description of the two models.

We are glad to hear that R2 thought the paper was "clearly written and will be an interesting topic to the readers of Biogeosciences." As well as, "In the discussion, what is missing in the current VDMs are thoroughly reviewed", which, as we listed above in our answer to concern #1, is the goal of this paper.

---

## Author Response (AR2)

Jan. 3rd, 2023
Dear Biogeosciences Editor and Reviewers,

We appreciate your review and comments on "Exploring the impacts of unprecedented climate extremes on forest ecosystems: hypotheses to guide modeling and experimental studies". Your time spent on this peer-review process is appreciated. Below is a point-by-point response to the reviews, and a description of how the manuscript has been revised accordingly.

Reviewer 3 agreed with the authors in the last revision, that since we do not intend this paper to be a classic model experiment, then "it is not necessary to do more simulation experiments and model validation since the simulations are just for illustrating the model behavior". However, reviewer 3 would like to see more of a review of "of VDMs processes and possible model development strategies, beyond the two models that are used to do simulation case studies."

We, the authors, agree with this comment that it is useful to include descriptions of model processes as represented in other VDMs (and some LSMs for comparison), and include recommendations for improvements, even though this will increase the length of the manuscript. Examples of the updated text describing additional VDMs are below.
However, we would like to be clear that the intention of this manuscript is not to be a classic review-**only** paper, and thus not structured as a typical review paper. We apologize if this was misleading in the previous author responses.
We are aiming for it to be a **perspectives paper, and a guidebook paper** for helping to guide the need for future experiments and/or monitoring data. Thus, we are "using a handful of novel model results to guide that discussion". The title of the paper is …."hypotheses to guide modeling and experimental studies", not a "review of modeling and experimental studies".
We would like to clarify that the original "research" aspects of this study were the new UCE model results, and specifically the integrated C-loss with sensitivity to different climate change treatments which has not previously been done with these models, and is a knowledge gap that was filled by this study.

In the revised manuscript we have included descriptions of additional VDMs, to help compare against the two main VDMs used here (ED2 and LPJ-GUESS) to evaluate carbon loss from UCEs. These models are listed below in the new table that was added to the supplemental materials.

**Table S1.** Terrestrial model full name, select characteristics, and associated references for the models listed throughout the manuscript.

| Model | Full Name | Type & Canopy | Dynamic Vegetation? | Plant Hydraulics? | References |
|---|---|---|---|---|---|
| CABLE | Community Atmosphere-Biosphere-Land Exchange | Big leaf; Single layer | No | No | Wang et al., (2011); |

| | | | | | |
|---|---|---|---|---|---|
| CABLE-POP | Community Atmosphere-Biosphere-Land Exchange - Population Orders Physiology | Cohort; Single layer | No | No | Haverd et al., (2018) |
| CLM5 | Community Land Model v5 | Big leaf; Single layer | No | Yes | Lawrence et al., (2019) |
| ED2-hydro | Ecosystem Demography v.2 - Hydro | Cohort; Multi-layer | Yes | Yes | Xu et al., (2016); Xu et al., (2021) |
| FATES | Functionally Assembled Terrestrial Ecosystem Simulator | Cohort; Multi-layer | Yes | No | Fisher et al., (2015) |
| FATES-HYDRO | Functionally Assembled Terrestrial Ecosystem Simulator - Hydro | Cohort; Multi-layer | Yes | Yes | Fang et al., (2022) |
| JSBACH4.0 | JSBACH v4 DGVM | Patch-tiling; Single layer | No | No | Nabel et al., (2020) |
| JULES | Joint UK Land Environment Simulator | Big leaf; Single layer | No | Yes | Eller et al., (2020) |
| LM3-PPA | Land Model v3 – Perfect Plasticity Approximation | Cohort; Multi-layer | Yes | No | Weng et al., (2015) |
| LPJ-GUESS | Lund-Potsdam-Jena General Ecosystem Simulator | Cohort; Multi-layer | Yes | No | Smith et al., (2001); Smith et al., (2014) |
| Noah-MP-PHS | Noah-Multiparameterization - Plant Hydraulics Scheme | Big leaf; Single layer | No | Yes | Li et al., (2021) |
| ORCHIDEE | ORganizing Carbon and Hydrology in Dynamic EcosystEms | Big leaf; Single layer | Yes | No | Krinner et al., (2005); Druel et al., (2019) |
| SEIB-DGVM | Spatially Explicit Individual-Based Dynamic Global Vegetation Model | Individual; Multi-layer | Yes | No | Sato et al., (2007) |

| TFSv.1-Hydro | Trait Forest Simulator v1 - Hyrdo | Individual; Multi-layer | No | Yes | Christoffersen et al., 2016 |
|---|---|---|---|---|---|

To make the paper more in-line as a perspectives paper, we will remove some instances that resemble a classical modeling experiment/comparison paper. For example, we removed the site description of the two sites from Section 2 of the manuscript to the Supplemental Material. However, comments from reviewer #1 was concerned about the lack of model validation compared to site level observations. Therefore, we kept the text in the manuscript on basic model validation to still address review #1 comments.

This paper was intended to describe current model mechanisms that could affect pre-drought resistance and post-drought recovery and suggest critical areas further research (see Table 1 and Table 3), including many processes such as hydrodynamics, mortality, allocation, community assemble, phenology, carbon storage, etc. Each of these mechanisms could be their own separate review paper, therefore we only included summaries and critical highlights of each.

In the updated manuscript, we refer the readers to "existing review papers of different VDM development, processes, and uncertainties can be found here: Fisher et al., (2018); Bonan (2019); Trugman et al., (2019); Hanbury-Brown et al. (2022); Bugmann and Seidl (2022); and specifically related to plant hydraulics see: Mencuccini et al., (2019); Anderegg and Venturas (2020)."

**Additional responses to specific reviewer comments:**

Key:
Gray text: comment from reviewers
Black text: response from the authors
Red text: edits or new text added to the manuscript

**Reviewer 3 expressed concerns about:**

1. **R3:** "Generally, I agree with the authors' argument that, as a review paper, it is not necessary to do more simulation experiments and model validation since the simulations are just for illustrating the model behavior. However, if it is a review paper, I think, it is necessary to include more detailed analysis of VDMs processes and possible model development strategies, beyond the two models that are used to do simulation case studies."

**Answer:** We agree with this comment that it would be useful to have more background of VDM processes and different model development strategies (beyond the two models used in the paper). To address this comment, we made substantial changes to Section 2 of the paper, which is now called: "Vegetation Demography Model (VDM) Strategies". We will start this section by talking more generally about the use and strategies of VDM application in investigating unknown future climate extremes, and as one update, provide more specific examples of which models use mechanistic hydrodynamics or

not. Since ED2 and LPJ-GUESS are the only models used to showcase the integrated carbon loss results, we retained a more detailed description of the capabilities of these two models (Section 2.1).

We also changed the naming of section 2.2 from "Modeling protocol" to "Modeling guide" to convey more of a potential guidebook option for future research.

In the Discussion section of the new manuscript, we will also provide examples from other VDMs. R3 suggested specific examples related to canopy dynamics, mortality, allocation, and community assemblage in their "specific comments" section, and we will update the text accordingly. Please see below for the new Discussion text that is included in the manuscript.

The beginning of Section 2 text has been updated to include the following:

"We argue that VDMs are well suited to address climate change impacts due to the inclusion of detailed process representation of dynamic plant growth, recruitment, and mortality, resulting in changes in abundance of different PFTs, as well as vertically stratified tree size- and age-class structured ecosystem demography. Community dynamics and age-/size-structure are emergent properties from competition for light, space, water, and nutrients, which dynamically and explicitly scale up from the tree, to stand, to ecosystem level. Within this characterization, VDMs also differ between each other and are set up in different configuration, allowing for various testing capabilities. For full names of each model listed below and references, see Table S1. For example, VDMs can aggregate and track the community level disturbance into either patch-tiling sampling (e.g., ED2, FATES, LM3-PPA, ORCHIDEE, JSBACH4.0) or statistical approximations (e.g., LPJ-GUESS, SEIB-DGVM, and CABLE-POP). VDMs could also vary in representing light competition within either multiple canopy layers (e.g., ED2, FATES, LM3-PPA, LPJ-GUESS, SEIB-DGVM) or in a single canopy (e.g., JSBACH4.0, ORCHIDEE, CABLE-POP).

Powell et al. (2013) compared multiple VDMs and LSMs to interpret ecosystem responses to long-term droughts in the Amazon and are informative when conducting model-data comparisons, but studies of the cascade of ecosystem responses and mortality to UCEs are lacking. In a cutting-edge area of development, new mechanistic implementation of plant competition for water and plant hydraulics in VDMs (i.e., hydrodynamics) are improving our understanding of plant-water relations and stresses within plants, such as with TFSv.1-Hydro (Christoffersen et al., 2016), ED2-hydro (Xu et al., 2016), and FATES-HYDRO (Ma et al., 2021; Fang et al., 2022). Compared to more simplistic representation of plant acquiring soil moisture not connected to plant physiology (e.g., LPJ-GUESS, LM3-PPA, CABLE-POP, SEIB-DGVM). For hydrodynamic representations in 'big-leaf' LSMs such as CLM5, JULES, and Noah-MP-PHS see Kennedy et al., (2019), Eller et al., (2020), and Li et al., (2021) respectively.

The discussion section provides a deeper investigation of model response to UCEs related to droughts. An exhaustive review of all VDMs, and all plant processes is too large to be done here. Existing review papers of different VDM development, processes, and uncertainties can be found here: Fisher et al., (2018); Bonan (2019); Trugman et al., (2019); Hanbury-Brown et al. (2022); Bugmann and Seidl 2022; and specifically related to plant hydraulics see: Mencuccini et al., (2019); Anderegg and Venturas (2020). We use LPJ-GUESS and ED2 as example VDMs in an initial guide framework to explore hypotheses around vegetation mortality and integrated

carbon loss from UCEs and climate change impacts, and highlight limiting model processes. Since field data needed to evaluate UCE responses are, by definition, unavailable, we do not perform model-data comparisons. Rather, we use the model results and conceptual framework as a road map to explore our hypotheses and illustrate their implications for ecosystem responses under UCEs, not historical drought events."

To account for the added text of describing additional VDMs, and to remove instances that resembled a more classic modeling experiment study. For example, we moved the description of the two sites from section 2.2 to the Supplemental Material.

2. **R3:** "There are many processes…related to the responses to the "unprecedented climate extremes (UCEs)". These processes, for example, may include physiological processes (photosynthesis, respiration, plant hydraulics, allocation, etc.), demographic processes (regeneration and mortality), community processes (e.g., competition and community compositional shifts), and soil organic matter decomposition and biogeochemical cycles. If the authors claim this is a review paper, it should be more comprehensive and insightful in discussing these processes than current version.

**Answer:** We agree that many plant processes are sensitivity to, and influence the response to UCEs. Thus, we have Table 3 that describes a summary of driving mechanisms (e.g., ecosystem or plant processes and state variables) that could be used to guide future research in manipulation experiments, data collection, and model development and testing, as related to furthering our understanding of UCE resistance and recovery. A critical look at these processes (such as phenology, plant hydraulics, carbon storage, allocation, soil water availability, functional diversity, etc.) emerged from the hypothetical drought simulations used in this study. We now see that our first paper submission did a poor job at highlighting Table 3 and Table 1, which described many of the ecological processes and mechanisms, and we have improved the manuscript by highlighting these evaluations.

We apologize if our previous author responses were misleading, because we did mean to claim or intend that this manuscript is a review-only paper. We are aiming for it to be a perspectives paper, and a guidebook paper for helping to guide the need for future modeling experiments and/or monitoring data.

At the beginning of the Discussion section, after this sentence "These nonlinearities may arise from multiple mechanisms that we begin to investigate here, including shifts in plant hydraulics or other functional traits, C allocation, phenology, and stand demography, all which vary among ecosystem types." We include this new sentence to make it clear to the reader that we provide suggestions for model mechanisms to further explore at line 559: "A critical look of driving model mechanisms, which emerged from the hypothetical drought simulations used here, are summarized in Table 3."

We also include this new sentence in the Introduction:

"Table 1 describes a summary of model mechanisms that affect pre-drought resistance and post-drought recovery and we suggest are critical areas further research."

Specifically related to leaf turnover and mortality, we also include this new text starting at line 582:

"Leaf loss is one component of total carbon turnover flux equations in terrestrial models, in addition to woody loss, fine-roots, and reproductive tissues. Having a better understanding of when extreme levels of phenological turnover contribute to stand-level mortality could be improved. Among other turnover hypothesis explored, Pugh et al. (2020) found that phenological turnover fluxes where just as important as mortality fluxes in driving forest turnover time in the VDMs: LPJ-GUESS, CABLE-POP, ORCHIDEE, but not the LSM JULES."

Specifically related to carbon allocation strategies, we also include this new text starting at line 671:

"Further eco-evolutionarily-based approaches such as optimal response or game-theoretic optimization, as well as entropy-based approaches are useful when wanting to simulate higher levels of complexity (reviewed in Franklin et al. 2012). With more frequent UCEs and if plants need to reduce water consumption, the optimal strategy of allocation between leaves and fine roots should change, therefore the goal functions (e.g., fitness proxy) in optimal response modeling can account for costs and benefits of allocation shifts between all organs (Franklin et al. 2009, 2012)."

Specifically related to community composition, we also include this new text starting at line 812:

"Due to VDMs being able to exhibit dynamic biogeography they are more useful at predicting shifts in community composition beyond LSMs capabilities. Further areas of advancement (described in Franklin et al. (2020)) is including models of natural selection, self-organization, and entropy maximization which can substantially improve community dynamic responses in varying environments such as UCEs. Eco-evolutionary optimality (EEO) theory can also help improve functional trait representation in global process-based models (reviewed in Harrison et al., 2021), through hypotheses in plant trait trade-offs and mechanistic links between processes such as resource demand, acquisition, and plant's competitiveness and survival; traits associated with high degrees of sensitivity in models. The power of prognostic VDMs to predict shifts in demography and community migration with climate change is large, but rarely is being constrained with plant-level EEO theory, and thus will likely need to use stand level competition and coexistence principles of how plants self-organize (Franklin et al. 2020)."

3. **R3:** "Many review comments (especially the Specific comments) are dismissed by simply arguing "this is a review paper" or never mentioned. I think a point-by-point response is still necessary."
**Answer:** We apologize if we missed some specific comments from previous reviewers and did not address their points adequately. Under the Interactive Discussion we

noticed that some comments were missing from the last review period, and we have added new author replies under the "reply to RC". We hope we have addressed any outstanding comments.

4. **R3**: "The authors should update accordingly by including the discussions of the key processes and generalize model development perspectives in the VDMs beyond these two models. I didn't see this kind of revisions in the revised manuscript"

    **Answer:** Similar to comment #1 above, we have updated the manuscript to include descriptions of other VDMs beyond the two models used here. We have also included more discussion of general VDM processes and model development that could be prioritized for understanding ecosystem response to UCEs.
    See comment #6 below for updates to leaf loss and mortality.
    See comment #7 below for updates to the role of plant hydraulics in other VDMs.

    Specifically related to carbon allocation strategies, we also include this new text starting at line 658:
    "Global scale terrestrial models are beginning to include optimal dynamic C allocation schemes, over fixed ratios, that account for concurrent environmental constraints on plants, such as water, and adjust allocation based on resource availability such as in LM3-PPA (Weng et al., 2015),"

    We would like to emphasize the integrated-c-loss and integrated-c-change due to climate change as a new methodology to explore in all VDMs, not just a one-time application used here. Therefore, we have introduced this approach earlier in the abstract.
    "As a result of nonlinear ecosystem responses to UCEs, that are qualitatively different from responses to milder extremes, we consider both biomass loss and recovery rates over time, by reporting a time-integrated carbon loss as a result of UCE, relative to the absence of drought."

5. **R3**: Concerns with linear vs. non-linear ecosystem response.

    **Answer:** We agree that complex biological life systems are not linear. The previous text (i.e., mentions of linear vs. nonlinear response) was not an appropriate description of what we were trying to convey, and has been corrected in our revised manuscript. The correct hypotheses that we were trying to test was comparing different degrees, or amplitudes, of nonlinearities, with carbon loss becoming more strongly nonlinear with increasing UCEs. Therefore, we removed text that described the null hypothesis as a linear relationship between carbon stock and drought, and instead are describing the null hypothesis and near-linear, and alternative hypothesis as different degrees of non-linearities.

In section 1.1 we updated the following text:

"Change in vegetation C stock is related to drought intensity and/or drought duration in a near-linear relationship (Fig. 1a, H0, null hypothesis), which has some observational support from annual and perennial grassland ecosystems, shrublands and savannas across the globe (Bai et al., 2008; Muldavin et al., 2008; Ruppert et al., 2015). We recognize that most ecological systems are nonlinear, thus alternatives to the null hypothesis are that biomass loss increases non-linearly with increased drought intensity"

6. **R3:** "LAI and its profile (Section 4.1.1 The role of phenology and phenological strategies prior to UCEs)". Lines 485~487: "ED2 predicted a very weak biomass loss at the time of UCEs (Fig. 2a), suggesting large-scale leaf loss is not a direct mechanism of plant mortality in ED2." This is a good place to explain the mechanisms of leaf loss, plant growth, and mortality in VDMs and weakness of current VDMs, by using ED2 as a case.

**Answer:** In an effort to make this more of a "perspectives" paper we took the reviewers advice and used ED2 as a case to explain the mechanisms of leaf loss and growth/mortality in VDMs.

We include the following new text starting at line 582: "Leaf loss is one component of total carbon turnover flux equations in terrestrial models, in addition to woody loss, fine-roots, and reproductive tissues. Having a better understanding of when extreme levels of phenological turnover contribute to stand-level mortality could be improved. Among other turnover hypothesis explored, Pugh et al. (2020) found that phenological turnover fluxes where just as important as mortality fluxes in driving forest turnover time in the VDMs: LPJ-GUESS, CABLE-POP, ORCHIDEE, but not the LSM JULES."

We did some word-smithing to the final sentence in section 4.1.1 to improve our point of highlighting areas for improvement in VDMs. We also added the final sentence about examples used in other VDMs (e.g., FATES model). It now reads:

"VDMs could be improved by better capturing different plant phenological responses to UCEs by better representing a range of leaf-level morphological and physiological characteristics relevant to plant-water relations such as leaf age, retention of young leaves even during extreme droughts, (Borchert et al., (2002)), and variation in hydraulic traits as a function of leaf habit (Vargas et al., (2021)) (Table 3). Two such examples are seen in the FATES model where the possibility for "trimming" the lowest leaf layer can occur when leaves are in negative carbon balance due to light limitation thus optimizing maintenance costs and carbon gain, as well as leaf age classifications providing variations in leaf productivity and turnover."

In section 4.1.1 we provide an example from another VDM (CABLE), stating that when "evaluating the simulated response to $CO_2$ fertilization, LAI was the largest source of variability in CABLE (Li et al., 2018)."

7. **R3:** Mortality (mentioned in 4.1.2 The role of plant hydraulics prior to UCEs).
In section "4.1.2 The role of plant hydraulics prior to UCEs", the authors mentioned different mortality settings in the two case models, but no details and no mentions of other models.
**Answer:** We agree that this section would be a good place to describe connections between mortality and role of plant hydraulics in other VDMs (other than ED2 and LPJ-GUESS). However, other reviewers have pointed out that our manuscript is already very long and an exhaustive review of plant hydraulics in more models would greatly increase the length of this paper. We would like to find a good balance, and briefly explain other models, but also point the readers to model evaluation manuscripts where they can find more information.

We include the following new text starting at line 618:
"Of the VDMs that are beginning to incorporate a continuum of hydrodynamics (e.g., ED2 (described in Methods 2.1 section) and FATES-HYDRO (Fang et al., 2022 based on Christoffersen et al., 2016), they are able to solve for transient water from soils to roots, through the plant and connect with transpiration demands. Therefore instead of the plant water stress function being based on soil water potentials, it is replaced with more realistic connections with leaf water potentials. Mortality is then caused by hydraulic failure via embolism controlled by the critical water potential ($P_{50}$) that leads to 50% loss of hydraulic conductivity. For advancements in tree level hydrodynamic modeling see the FETCH3 model (Silva et al., 2022), for justification for plant hydrodynamics in conjunction with multi-layer vertical canopy profiles see Bonan et al., (2021)."

In Section 2 of the updated manuscript, we also include the following text on plant hydraulics in other models:
"In a cutting-edge area of development, new mechanistic implementation of plant competition for water and plant hydraulics in VDMs (i.e., hydrodynamics) are improving our understanding of plant-water relations and stresses within plants, such as with TFSv.1-Hydro (Christoffersen et al., 2016), ED2-hydro (Xu et al., 2016), FATES-HYDRO (Ma et al., 2021; Fang et al., 2022). Compared to more simplistic representation of plant acquiring soil moisture not connected to plant physiology (e.g., LPJ-GUESS, LM3-PPA, CABLE-POP, SEIB-DGVM). For hydrodynamic representations in 'big-leaf' LSMs such as CLM5, JULES, and Noah-MP-PHS see Kennedy et al., (2019), Eller et al., (2020), and Li et al., (2021) respectively."

8. **R3:** Allocation (4.1.3. The role of carbon allocation prior to UCEs)
The authors attributed shedding leaves as a strategy of allocation. I think it more close to a strategy of reducing water consumption. With more frequent UCEs, the optimal strategy of allocation between leaves and fine roots should change. How it will shift is an interesting topic that may need to discuss in this section. See a review paper: (Franklin et al., 2012)
**Answer:** We thank the reviewer for this point, and providing the Franklin paper on 'Modeling Carbon Allocation'. We have updated the text to point out the strategy of changing allocation between leaves and fine root under water stress, and the need to reduce water consumption. We also included the Franklin et al., 2012 reference in Table 1.

We include the following new text starting at line 671:

"Further eco-evolutionarily-based approaches such as optimal response or game-theoretic optimization, as well as entropy-based approaches are useful when wanting to simulate higher levels of complexity (reviewed in Franklin et al. 2012). With more frequent UCEs and the need for plants to reduce water consumption, a shift in the optimal strategy of allocation between leaves and fine roots should change. The goal functions (e.g., fitness proxy) used in optimal response modeling can account for these shifts in costs and benefits of allocation between all organs (Franklin et al. 2009, 2012)."

Text that was related to "shedding leaves" (as pointed out by the reviewer) was located in the plant carbon storage section 4.1.4 (storage of non-structural carbohydrates (NSCs)) where we describe one way for maintenance of NSCs is that trees can resorb a fraction of leaf C during leaf shedding in ED2. We also mentioned that interactions with rising atmospheric $CO_2$, NSCs, and leaf shedding "needs to be further explored".

9. **R3:** In line 464~466 "shifts in plant hydraulics or other functional traits, C allocation, phenology, and stand demography": The processes mentioned in this sentence are for one PFT. However, for VDMs, the shifts also include structural and compositional changes.
   **Answer:** We thank the reviewer for pointing out that VDMs also include processes like structural and compositional changes across varying community assemblages (i.e., not just one PFT), and this sentence has been updated accordingly.

   We include the following new text starting at line 556:
   "…including shifts in plant hydraulics or other functional traits, C allocation, phenology, stand size-structure and/or age demography, and compositional changes, all which vary among ecosystem types. A critical look of driving model mechanisms, which emerged from the hypothetical drought simulations used here, are summarized in Table 3."

10. **R3:** I suggest adding a section discussing about the community level reorganization due to UCEs or re-organize the section "4.2.3 The role of stand demography post-UCEs" for this. Please refer to the papers (Mencuccini et al., 2019; Franklin et al., 2020; Harrison et al., 2021).
    **Answer:** To address this comment we have added text to include the optimality approaches described in Franklin et al., 2020, Harrison et al., 2021. (A note that the Mencuccini et al. 2019 was a review of water fluxes in plants and was included in the plant hydraulics section).
    The research area of optimality approaches in modeling (i.e., eco-evolutionary optimality (EEO), evolutionarily stable strategy (ESS), entropy-based approaches, self-organization) is *very large*, and could be its own separate paper(s). We have tried to

highlight how optimality hypotheses can be used to describe community level compositional changes due to UCEs, but we cannot go into full description of each of these concepts here.

We have re-organized the section "4.2.3 The role of stand demography post-UCEs" to include these new topics. We also included the Franklin et al., 2020 reference in Table 1 and Table 3.

We include the following new text starting at line 812:

"Due to VDMs being able to exhibit dynamic biogeography they are more useful at predicting shifts in community composition beyond LSMs capabilities. Further areas of advancement (described in Franklin et al. (2020)) is including models of natural selection, self-organization, and entropy maximization which can substantially improve community dynamic responses in varying environments such as UCEs. Eco-evolutionary optimality (EEO) theory can also help improve functional trait representation in global process-based models (reviewed in Harrison et al., 2021), through hypotheses in plant trait trade-offs and mechanistic links between processes such as resource demand, acquisition, and plant's competitiveness and survival; traits associated with high degrees of sensitivity in models. The power of prognostic VDMs to predict shifts in demography and community migration with climate change is large, but rarely is being constrained with plant-level EEO theory, and thus will likely need to use stand level competition and coexistence principles of how plants self-organize (Franklin et al. 2020)."

11. **R3:** Successional patterns (4.2.4 The role of functional trait diversity & plant hydraulics post-UCEs): Lines 680~681: "disturbance, competition will likely shift the plant community towards one that is composed of opportunistic, fast-growing pioneer tree species". I think this is an issue of recovery trajectory. If the climate conditions are the same and the PFTs are available, the vegetation should recover to its pre-UCE states, depending on model setting.

**Answer:** We thank the review for pointing this out. The references associated with this sentence are from field experiments, not model results. We have updated the text to say "Higher" disturbances rates (implying repeated, more extreme disturbances) will likely shift the recovery trajectory of the plant community.

The sentence has been updated to: "In field experiments, higher disturbance rates have shifted the recovery trajectory and competition of the plant community towards one that is composed of opportunistic, fast-growing pioneer tree species, grasses"

12. **R3:** Lines 685~691: For the issue "ED2 exhibited a strong recovery in the evergreen PFT as well (over two other deciduous PFT types), inconsistent with the above literature", the authors assumed the reason is in the response to drought (i.e., plant hydraulics issue). It could be the nitrogen issue, related to the feedback between leaf traits (LMA) and litter decomposition. Please see (Aerts, 1995, 1999)

**Answer:** We agree that nitrogen cycling feedbacks between leaf traits and litter decomposition could be an explanation for the evergreen recovery. However,

biogeochemical cycling was not investigated in this study, as we tried to focus more on hydraulic traits and processes.  Regardless, we would like to point out to the reader that differences in nitrogen demands, etc. should be considered.

We include the following new text starting at line 839:
"Nitrogen cycling feedbacks were not investigated here, but could also be an explanation for a strong evergreen PFT recovery."

**Reviewer 4 expressed concerns about:**
1. **R4:** A component of the inter-modeling comparison.

**R4:** I am hard to understand what is happening in the models when I see the simulation results in Figure 3. I expect higher environmental CO2 (eCO2) to deliver higher tolerance to drought for plants. Therefore, higher eCO2 would deliver higher carbon storage in vegetation under drought conditions. But, runs with only-increased-eCO2 frequently result in lower vegetation carbon than runs with warming-plus-increased-eCO2 (Fig. 3 a,e,f,g,h,i). In lines 412-413, the authors also mentioned that "losses are exacerbated when accompanied with warming and even with eCO2, with 800 ppm having a more detrimental impact than 600 ppm (Fig. 3a-c)."

The authors already notice this problem because they state, "The VDM simulations suggest that the combination of elevated warming and eCO2 will exacerbate consequences of UCEs by reductions in both C stocks and post-drought biomass recovery speeds" (Lines 715-716). Although, I do not want to encourage authors to increase the length of the discussion, as it is already too long. But, the authors should state the possible reason(s) for this mismatch.

**Answer:** We appreciate the reviewer highlighting this point about elevated $CO_2$ climate change results.
The reviewer points out that the 'only $CO_2$ fertilization' treatment should give higher carbon storage, and higher biomass.  But, in 5 of the 12 results in Figure 3 (Fig. 3 a,e,f,g,h,i) the only-increased-$eCO_2$ results in lower vegetation carbon, or a more negative integrated C change (blue lines), compared to runs with warming-plus-increased-eCO2 (red lines).

We believe a main explanation for this is related to the structural overshoot response with $CO_2$ fertilization that we describe in our conceptual Figure 1. There is an amplification of vegetation biomass due to eCO2, and either favorable historical conditions, or recovery from smaller disturbances (or both). Our argument is that when there is are **extreme droughts (or mortality events)** the loses are compounded since there is previously more biomass stock in the system, and there ends up being a negative C change from baseline, or the mortality overshoot.  This theory is also confirmed when we see that the highest eCO2 (800ppm, purple lines) results in the lowest carbon change, when combined with the highest drought intensities (Fig. 3g,h,i,j).
We will make this clear in the revised manuscript.

In Section 4.1.3 "The role of carbon allocation prior to UCEs" we discuss this connection between the structural overshoot with $eCO_2$ and then the mortality overshoot with UCEs. "Mortality overshoot, as a result of structural overshoot, could be an explanation for the negative integrated-C-change (i.e., C loss) in the majority of $eCO_2$-only simulations (18 out of 24 scenarios; Table 2)."

We have updated the line in the summary to include the explanation of structural overshoot with $eCO_2$ and increased competition for resources at line 870:
"The VDM simulations suggest that the combination of elevated warming and potential structural overshoot from $eCO_2$ (or inaccurate representation in NSCs allocation/usage priority) will exacerbate consequences of UCEs by reductions in both C stocks and post-drought biomass recovery speeds (Fig. 3)."

We have also updated the sentence at line 500 to only include Fig. 3b-c, not Fig. 3a-c. And we incorrectly listed the $CO_2$ ppm. It should read the other way around that the lower 600 ppm led to a more detrimental impact in carbon loss (in ED2) compared to the higher 800 ppm $CO_2$. This better matches the reviewers' comments that higher $eCO_2$ should deliver more carbon to the vegetation.
"losses are exacerbated when accompanied with warming, even with $eCO_2$, with 600 ppm having a more detrimental impact than the more elevated 800 ppm (Fig. 3b-c)."

---

## Author Response (AR3)

April 3rd, 2023
Dear Biogeosciences Editor and Reviewers,

We appreciate your review and comments on "Exploring the impacts of unprecedented climate extremes on forest ecosystems: hypotheses to guide modeling and experimental studies". Your time spent on this peer-review process is appreciated. Below is a point-by-point response to the reviews, and a description of how the manuscript has been revised accordingly.

Key:
Gray text: comment from reviewers (e.g., RC)
**Black text**: response from the authors
Red text: edits or new text added to the manuscript

Comments from Referee #4: "The corresponding author have addressed my concerns and I'm happy with them."
**Author response**: We are pleased to hear that this reviewer was happy with the latest submitted revisions, and was happy with the changes.

Comments from BG Editor:

RC: Your overall aim, to provide a perspectives paper, is still hampered by a focus on details on the two individual models, making it hard for a reader to see the general points you are trying to make behind the results of the two models. Furthermore I find that in some places the framework and pathways to future research are not very clear.
**Author response**:
- Thank you for this point that there is still too much focus on individual model results, when the models are intended to be a guide to investigate unknowns and areas for improvements. At the beginning of the results section, we removed the text that validated the models to observations, and moved these results to the supplements.
- We tried to also reduce the emphasis regarding the diverging responses between the individual models that one happened to be sensitive to drought intensity, while the other model was sensitive to drought duration. We do not want to claim that one response pattern (intensity vs. duration) is "correct" over the other, but instead there are a lot of uncertainties with representing future extremes (conveyed by the increasing amplitude and shading in panel 1d in our conceptual Figure 1).

RC: Regarding H1: Why would you assume that the response to drought would be linear? We know that stomatal responses to soil moisture are not, so why should the response to extremes be linear?
**Author response**:
- We agree that response to droughts are not linear, and this linear hypothesis option is not necessary. We have removed any instances of a linear response, and only describe UCE impacts as varying degrees of non-linearity and threshold responses.
- Figure 1 has also been updated. We removed the null hypothesis that responses might be linear, or near-linear.

RC: Most of the results shown and mechanisms discussed refer to UCEs in water availability and drought. I think it's important to point this out in Abstract, Intro and Conclusions, because the manuscript does not address concepts applicable to for instance heat stress, cold snaps etc.

**Author response**:

- We updated the manuscript to more clearly point out in the abstract, introduction, and conclusions that this paper is *only* discussing UCEs related to water availability and drought, not other extremes.
- Line 41: "Here, we present a road map of how two dynamic vegetation demographic models (VDMs) can be used to investigate hypotheses surrounding ecosystem responses to one type of UCE: unprecedented droughts."
- Line 138: "While a variety of UCE-linked biophysical tree disturbance processes (e.g., fire, wind, insect outbreaks) can drive nonlinear ecosystem responses, we focus specifically on extreme droughts, which have important impacts on many ecosystems around the world (e.g. Frank et al., 2015, IPCC 2021)."

RC: You suggest that you develop an iterative framework. However, it remains unclear, in which respect or how the framework is iterative, which I read to refer to a loop of experiment - model development & testing - experiment -model development and testing and so forth.

**Author response**:

- We updated the manuscript to make sure we are not claim that we have developed an iterative framework currently in this manuscript, especially since we only include modeling testing and do not introduce experiments here. But that an iterative framework is suggested for future work, and is a useful tool for understanding responses to extremes.
- Line 146: "This study can help guide how the scientific community can iteratively address these questions through future experiments and modeling studies."

RC: Why are VDMs needed for this exercise? Could simpler models not also show similar behaviour? Briefly motivate why these two models were chosen to exemplify your framework.

**Author response**:

- We intentionally choose to use VDM for this exercise of investigating extremes due to their representation of ecosystem structure, demography, and capturing competitive responses from disturbances. Simpler models fail to mechanistically represent mortality, recruitment, and disturbance – each of which influences biomass turnover and carbon (C) allocation (Friend et al., 2014) – and thus non-VDMs are limited in their ability to realistically forecast ecosystem responses to anomalous environmental conditions like UCEs (Fisher et al., 2018).
- This is further described and justified on page 4 in the introduction, starting at line 110.

RC: Since your paper is meant to be a perspectives paper, a brief elaboration of why you chose these two sites out of the many possible as representative would be warranted.

What are you trying to demonstrate with these two different sites? Why not only use one, if the disturbance simulated anyway do not correspond to the experiments at the site? I do not get the sense that the use of more than one site adds a lot of information to the manuscript that is accessible to the reader and serves the purpose of a perspectives paper. A little more explanation would be beneficial here.

**Author response**:

- In Section 2.2 of the manuscript, we have the following text describing the two site selections. "To exemplify how VDMs can be tools to explore new hypotheses related to UCEs we applied the models at two field sites, that were chosen due to being extensively studied and the models used here have already been run at these sites and previously benchmarked against field data (see Xu et al., 2016; Medlyn et al., 2016; Medvigy et al., 2019 for model-data validation)….. In addition, the two sites span a range of vegetation types and are in warm, seasonally dry climates that are more likely to experience droughts in the future (Allen et al., 2017)."

RC: Since your paper is meant to be a perspectives paper, the first paragraph in the Results section can be moved to the supplement. Since no data is shown to evaluate the response of the models in the light of their initial performance (and this is not a MIP-paper), I feel this paragraph is a distraction.

**Author response:**

- We agree with the reviewer, and these results have been moved to the supplements in the revision. A previous reviewer requested more site level validation and comparison to observations, so we included this observational data. However, we agree and think it can be in the supplements.

RC: While I understand that a in-depth discussion of the different responses of the two models is beyond this paper, it is somewhat dissatisfying that you emphasise the difference between the drought intensity and duration response of the two models without giving a clear hint as to the source of this. Either there is an explainable difference that can be summarised briefly in Section 4, or it should not be highlighted, because it remains unclear why this difference occurs. At least one could re-iterate the key differences between the models that contribute to this behaviour, rather than simply listing all possible causes that may or may not be included in the models.

**Author response**:

- We thank the reviewer for understanding that an "in-depth discussion of the different responses of the two models is beyond this paper" and also not the scope of the paper to be a model intercomparison. We've re-wrote parts of the abstract and conclusions to try and de-emphasize the diverging responses between the models that one happened to be sensitive to drought intensity, while the other model was sensitive to drought duration. We do not want to claim that one response (intensity vs. duration) is "correct" over the other, but instead there are a lot of uncertainties with representing future extremes (conveyed by the increasing amplitude and shading in panel 1d in our conceptual Figure 1).
- We instead try to emphasize that UCE impacts can be extremely variable, unknown interactions, and we still have some work to do in order to narrow down the potential plausible scenarios until

we have better grasp on processes surrounding plant hydraulics and stress, carbon allocation, large tree mortality, diverse community composition, etc.

- We updated the abstract to include these sentences: "The severity and patterns in biomass losses differed sustainably between well tested models. For example, biomass loss could be sensitive to either drought duration or drought intensity depending on the model approach. This is due to the models having different, but also plausible representations of processes and interactions, highlighting the complicated interactions and variability of UCE impacts still needed to be narrowed down in models."

Minor comments:

Abstract:

RC: L 41: remove reference to _two_ models. Why would your roadmap only apply to these two, and what interest would a reader of BG have if this manuscript is only relevant for these few model developers?
**Author response**:
Done. We have removed the reference to "two" models.

RC: L50: Complicated sentence, consider rewriting.
**Author response:**
Done. We have rearranged, and re-written, the sentences from line 50-54 to make more sense.

RC: L61: It would be preferable if the abstract would focus more on what this framework contains than individual model results that are only a guidance to develop the framework
**Author response:**
This is a good point, and we agree with the reviewer. Since this was not intended to be an in-depth model comparison paper, we removed one of the two times we mentioned in the abstract that the models reported either a stronger response to drought intensity vs. duration. Instead, we would like to emphasize that these different results are both plausible based on our current understanding of processes in these models. I.e., we can't narrow down the mechanisms (listed in the discussion) until more experiments can be done to explore the unknown UCE interactions/impacts.

RC: L104: drop reference to IPCC here. IPCC only assess existing science. CMIP6 is the framework in which these model runs were done. Ciais 2013 is not a recent IPCC assessment. Consider citing IPCC AR6 or IPCC SRCCL here.
**Author response:**
- Done.

RC: L348: Integrated carbon loss. I am struggling with the term, because it's a complicated mixture of intensity and duration of the response. A loss should simply have a unit of

kg/m2, but this is more a severity-of-impact index, and I recommend to call it that (or something comparable. Note that the unit varies in text and tables between kg C m-2 yr and kg C m-2 yr-1. Please check and clarify.

**Author response:**

- We understand that there might be some confusion around the integrated carbon loss term, because as the reviewer points out it's a combination of disturbance intensity level, and the duration of the recovery period, integrated over a time period. Not just a one-time carbon loss. Therefore, we have changed the name from integrated-carbon-loss to "severity-drought index" when reporting the carbon loss from drought events only. And changed the name from integrated-carbon-change to "severity-climate index" when quantifying the difference of the effects of climate change plus droughts compared to just drought alone.

- We have also corrected the discrepancy in units to always be kg C m-2 yr, not per year.

- We updated Figures 1-3 so that the axes now have the correct metric term of either severity-drought index, or severity-climate index (kg C $m^{-2}$ yr). The figure captions have also been updated to try and explain this severity metric more clearly.

RC: L456: Use of SO and MO not clear here, please re-iterate

**Author response:**

- We agree that the beginning of this sentence was not clear, and also probably not needed. We updated the sentence to go straight into explaining the VDM model responses with regards to SO and MO patterns.

- Line 449: "When comparing VDM responses to increasing drought severity and its interactions with warming and $eCO_2$ (related to conceptual Fig. 1d), ED2 showed a more consistent MO response during UCEs and with additional warming and $eCO_2$ ....."

RC: L789: Check grammar: does one "do lists"?

- Corrected.

RC: Table 3/3 unclear what "ratio-based optimality partitioning theory is meant to refer to.

**Author response**:

- In term of carbon allocation strategy in plants, ratio-based optimal partitioning theory is allocation to plant organs based on the most limiting resources (McCarthy and Enquist, 2007), which is described in the text on line 574. Based on other reviewer comments we have suggested different optimality theories to potentially explore in models as ways to decrease the variability and unknowns in model results.

RC: Figure 1: The shading in 1d are non-trivial, but also not well explained. Please clarify in figure legend or text.

**Author response**:

- Thank you for pointing this out about Figure 1d. We have updated the methods text (line 206) to include: "Additionally, more climatic variability from unprecedented $eCO_2$ levels and warming will lead to unknowns in how ecosystems are affected in the future (i.e., the widening, and downward shape of the shaded areas compared to historical, Fig. 1d)."

We updated the figure caption to include: "We conceptualize how oscillations between SOs and MOs could be amplified and the widening of the shaded areas represents increased variability in how unprecedented eCO$_2$ levels and temperatures will affect ecosystems in the future compared to historical."

RC: Figure 4-5 are only referred to once in the discussion. I don't really see the need to include them in the main text

**Author response:**

- Figures 4-5 are used to help explain the wide range of variability when trying to represent demographic ecosystem response to UCEs. These figures help to make our point that there are different potential patterns that still need to be investigated when dealing with complex extremes, and which specific responses need to be explored more (i.e., disagreement between models) or not explored (i.e., better agreement between models). For example, both models indicate that large trees will be impacted the most by extreme droughts, agreeing with observations (reviewed in Discussion). But the models disagree on the recovery of different species composition. We hope because of these points that Figure 4-5 can remain in the main text.